# Parameter calibration and stomatal conductance formulation comparison for boreal forests with adaptive population importance sampler in the land surface model JSBACH

Jarmo Mäkelä[1], Jürgen Knauer[2], Mika Aurela[1], Andrew Black[3], Martin Heimann[2], Hideki Kobayashi[4], Annalea Lohila[1,5], Ivan Mammarella[5], Hank Margolis[6], Tiina Markkanen[1], Jouni Susiluoto[1,7], Tea Thum[2], Toni Viskari[1], Sönke Zaehle[2], and Tuula Aalto[1]

[1]Finnish Meteorological Institute, P.O. Box 503, 00101 Helsinki, Finland
[2]Max Planck Institute for Biogeochemistry, 07745 Jena, Germany
[3]University of British Columbia, Canada
[4]Institute of Arctic Climate and Environment Change Research, Japan Agency for Marine-Earth Science and Technology, Japan
[5]Institute for Atmospheric and Earth System Research / Physics, P.O. Box 48, Faculty of Science, FI-00014 University of Helsinki, Finland
[6]Department of Forest Sciences, University of Quebec, Canada
[7]School of Engineering Science, Lappeenranta–Lahti University of Technology, P.O. Box 20, FI-53851 Lappeenranta, Finland
**Correspondence:** Jarmo Mäkelä (jarmo.makela@fmi.fi)

**Abstract.** We calibrated the JSBACH model with six different stomatal conductance formulations using measurements from 10 FLUXNET coniferous evergreen sites in the Boreal zone. The parameter posterior distributions were generated by the adaptive population importance sampler (APIS), then the optimal values were estimated by a simple stochastic optimisation algorithm. The model was constrained with in-situ observations of evapotranspiration (ET) and gross primary production (GPP). We identified the key parameters in the calibration process. These parameters control the soil moisture stress function and the overall rate of carbon fixation.

The JSBACH model was also modified to use a delayed effect of temperature for photosynthetic activity in spring. This modification enabled the model to correctly reproduce the springtime increase in GPP for all conifer sites used in this study. Overall, the calibration and model modifications improved the coefficient of determination and the model bias for GPP with all stomatal conductance formulations. However, only the coefficient of determination was clearly improved for ET. The optimisation resulted in best performance by the Bethy, Ball-Berry and the Friend and Kiang stomatal conductance models.

We also optimised the model during a drought event in a Finnish Scots pine forest site. This optimisation improved the model behaviour, but resulted in significant changes to the parameter values except for the unified stomatal optimisation model (USO). Interestingly, the USO model demonstrated the best performance during this event.

# 1 Introduction

Plants exchange carbon dioxide ($CO_2$) and water vapour ($H_2O$) with the atmosphere. Sufficient soil water, irradiance and adequate temperature are required to maintain the exchange rates during the growing season. Disturbances in these conditions such as drought, cold temperature or low radiation cause the plants to respond to the environmental stress via stomatal closure and the decrease in photosynthesis and transpiration (Lagergren and Lindroth, 2002; Mäkelä et al., 2004; Gao et al., 2017). The capability of plants to recover from such events depends on species and their adaptation to site conditions (Kozlowski and Pallardy, 2002). Stress is part of the normal annual cycle of the plants, but occasionally it may exceed the limits of recovery.

Soil water deficit and high water vapour pressure deficit can result in suppressed plant transpiration (Bréda et al., 1993; Kropp et al., 2017). Globally, soil drought has been recognised as one of the main limiting factors for plant photosynthesis (Nemani et al., 2003) and boreal forests are known to occasionally suffer from soil drought (Muukkonen et al., 2015; Gao et al., 2016). The recovery of photosynthetic capacity in spring has been connected to temperature history, and to frequency of severe night frosts (Bergh et al., 1998; Bergh and Linder, 1999) that can reverse the recovery. Understanding, and correctly modelling, these phenomena are especially important for boreal forests (Bonan, 2008) under changing environmental conditions.

Ecosystem and land surface models, describing the plant photosynthesis, transpiration and soil hydrology related processes, usually include descriptions and parameterisations for various stress effects. These parameters often lack a theoretical foundation (Gao et al., 2002; Medlyn et al., 2011) and descriptions of vegetation drought response and phenology have been recognized to need better formulations and design (Richardson et al., 2012; Powell et al., 2013; Xu et al., 2013; Medlyn et al., 2016). These deficiencies restrict a model's predictive capability under changing environmental conditions, and call for specific parameterisations for different plant types and vegetation zones.

Stomatal conductance models describe the pathway of $CO_2$ and water through the leaf stomata by an electric circuit analogy (Nobel, 1999). The variations in stomatal opening and mesophyll structure are interpreted as resistances to water flow and the process is idealised via generalised parameterisation. Stomatal conductance models mainly differ in their choice of variables driving the stomatal closure, and their performance has been recently assessed in modelling studies by e.g. Egea et al. (2011); Knauer et al. (2015); Franks et al. (2018). However, it can be hypothesised that the choice of the stomatal conductance model affects the ecosystem model parameters more broadly as the stomatal conductance formulations vary in their responses to the different conditions. A holistic assessment of the performance of the stomatal conductance models together with ecosystem model parameter optimisation has been missing.

In many other studies, where the aim has been to optimise land surface model parameters, the optimisation is based on estimating the gradient of the cost function: Knorr et al. (2010) for JSBACH, Kuppel et al. (2012); Peylin et al. (2016) for ORCHIDEE and Raoult and Luke (2016) for JULES. Gradient-based methods are faster than Markov chain Monte Carlo (MCMC) methods as they strongly steer the sampling process to reach a minimum in the cost function (see e.g. Gelman et al., 2013). This approach also enables a more indefinite setting of parameter ranges (limits for acceptable parameter values) when compared to methods that sample the full parameter space. However, they are prone to get stuck in local minima, especially when the dimensionality of the parameter space increases. In the last few years, similar parameter estimations have also been

done for CLM by Post et al. (2017) using the DREAM$_{(zs)}$ (MCMC) algorithm with multiple chains, and for JULES by Iwema et al. (2017) with the BORG algorithm that employs multiple optimisation algorithms simultaneously. The DREAM algorithm is fully iterative, which limits the number of parallel processes to the number of parallel chains in use (when we do not account for the possibility of the model parallelisation that can be substantial). The applicability of the BORG algorithm is dependent on the algorithms in use and the expertise of the user (to choose the right algorithms etc.).

APIS is a Monte Carlo (MC) method that can be run iteratively as presented by Martino et al. (2015) but it is also straightforward to parallelise, since all samples prior to each adaptation (in our simulations 2000 draws) can be drawn and estimated simultaneously. This latter feature is useful to decrease the amount of real time required to run the algorithm when computer resources are not the limiting factor – APIS requires considerably fewer sequential estimates than typical Markov chain methods. In the iterative mode, automatic stopping rules can be easily implemented to indicate when additional samples are not required to improve the estimates. The APIS algorithm samples the full parameter space (as do MCMC methods) and can utilise a mixture of parameter prior distributions. Therefore, APIS can estimate complicated multidimensional probability distributions with relative ease. These aspects make APIS an attractive alternative to the other sampling and optimisation methods mentioned above.

In this study we apply the land surface model JSBACH for 10 boreal coniferous evergreen forest eddy covariance sites to examine the performance of different stomatal conductance models, and their effect on calibrated parameters related to photosynthesis, phenology and hydrology. First, we utilise APIS to sample the full parameter space with the different stomatal conductance formulations and to locate different modes of the target distributions (peaks of high probability). Second, using the distributions generated by APIS as the prior distributions, we optimise the parameters using a simple stochastic optimisation method. Finally, we assess the inter-site variability and the robustness of the calibrated parameters together with different stomatal conductance formulations. Optimised parameters for a specific drought are also investigated and compared with the parameters for the general optimisation.

## 2   Materials and methods

We will next introduce the measurement sites, followed by the model and modifications made to it. Afterwards we will give a general overview of the simulations as well as the sampling process, the algorithms and methods used to analyse the results.

### 2.1   Sites and measurements

We use data from 10 FLUXNET (doi:10.17616/R36K9X) sites characterised as coniferous evergreen forests. Site descriptions with appropriate references are provided in Table 1. The site-level half-hourly eddy-covariance (EC) measurements were quality checked and gap-filled when needed to produce continuous half-hourly and daily time series. The gap-filled and low-quality (based on FLUXNET data quality flags) measurements were masked, and the daily aggregates (usually means) were accepted as part of the calibration process if at least 60% of the values between 4:00 and 20:00 (i.e. daytime measurements)

for that day were unmasked. The daily aggregates of ET and GPP were used to calibrate and validate the model, whereas the half-hourly data were used as climate forcing (as explained later in Section 2.4.

Based on the quality and quantity of their respective measurements, the sites were divided into calibration and validation sites. Essentially, if we have enough data from a site, it is used for both calibration and validation purposes. We required the site to have at least eight years of measurements, where the first five were used for calibration, and the consecutive three for validation. Otherwise we used the site only for a three year validation. The FLUXNET datasets were missing both the long- and shortwave radiation for the two Russian sites, Fyodorkovskoye (RU-Fyo) and Zotino (RU-Zot). These were generated from ERA Interim data. The soil types of all of these sites can mostly be identified as mineral soils with varying sand, clay and peat contents. Fyodorovskoye and Poker Flat (US-Prr) are natural peatlands and Lettosuo (FI-Let) is a drained peatland site.

The measurement error in the EC flux data were separated into systematic and random errors. The main systematic errors (density fluctuations, high-frequency losses, calibration issues) were taken into account as part of the post-processing of the data, and the random errors tend to dominate the uncertainty of the instantaneous fluxes. The random error is often assumed Gaussian but can be more accurately approximated by a symmetric exponential distribution (Richardson et al., 2006). It increases linearly with the magnitude of the flux, with a standard deviation typically less than 20% of the flux (Richardson et al., 2008; Rannik et al., 2016). Our treatment of the measurement (and model) errors is explained in Section 2.9.

**Table 1.** Descriptions for the sites used in this study sorted by their FLUXNET identifier. The first six sites are used for both calibration and validation purposes, with the first five years of each site used for calibration. The last three years as well as the last four sites are used for validation only. The reported elevation is in meters above sea level, LAI is the one-sided leaf area index and the average stand age is in years, along with average annual precipitation (P) in mm and temperature (T) in degrees Celsius.

| Site id | lat | lon | elev. | dom. species | LAI | age | P | T | years | reference |
|---------|------|---------|-------|--------------------|-----|-----|-----|------|-----------|--------------------------|
| CA-Obs | 53.99 | -105.12 | 629 | *Picea mariana* | 3.8 | 135 | 406 | 0.8 | 1999–2006 | Chen et al. (2006) |
| CA-Qfo | 49.69 | -74.34 | 382 | *Picea mariana* | 3.7 | 112 | 962 | -0.4 | 2003–2010 | Chen et al. (2006) |
| FI-Hyy | 61.85 | 24.29 | 180 | *Pinus sylvestris* | 3.5 | 45 | 709 | 2.9 | 1999–2006 | Kolari et al. (2009) |
| FI-Ken | 67.99 | 24.24 | 337 | *Picea abies* | 2.1 | 100 | 484 | 0.4 | 2003–2010 | Aurela et al. (2015) |
| FI-Sod | 67.36 | 26.64 | 179 | *Pinus sylvestris* | 1.7 | 150 | 527 | -0.4 | 2001–2008 | Thum et al. (2007) |
| RU-Fyo | 56.45 | 32.90 | 265 | *Picea abies* | 4.5 | 200 | 711 | 3.9 | 2002–2009 | Launiainen et al. (2016) |
| CA-Ojp | 53.92 | -104.69 | 579 | *Pinus banksiana* | 2.6 | 100 | 431 | 0.1 | 2004–2006 | Chen et al. (2006) |
| FI-Let | 60.64 | 23.96 | 119 | *Pinus sylvestris* | 6.0 | 40 | 627 | 4.6 | 2010–2012 | Launiainen et al. (2016) |
| RU-Zot | 60.80 | 89.35 | 121 | *Pinus sylvestris* | 1.5 | 215 | 493 | -3.3 | 2002–2004 | Kelliher et al. (1998) |
| US-Prr | 65.12 | -147.49 | 210 | *Picea mariana* | 0.7 | 72 | 275 | -2.0 | 2011–2013 | Ikawa et al. (2015) |

## 2.2 The JSBACH model

JSBACH (Kaminski et al., 2013) is a process-based ecosystem model and the land surface component of the Earth System model of the Max Planck Institute for Meteorology (MPI-ESM). We ran JSBACH offline using meteorological measurements

from the flux towers to force the model. Implications of this one-way coupling with the atmosphere include lack of feedback from the surface energy balance to the atmosphere, i.e. latent and sensible heat fluxes and surface thermal radiation do not directly affect prescribed air temperature or humidity. Similarly, the feedback of the surface to the vertical transfer coefficients within the atmospheric surface layer is missing as the wind speed that drives mixing is prescribed. Furthermore, since we use

site level data (each site is represented as a single grid point), the grid resolution does not affect the results.

We focus only on the most essential parts of JSBACH relating to our work. A more complete model description with details on e.g. soil heat transfer, water balance and coupling to the atmosphere can be found in Roeckner et al. (2003), whereas Raddatz et al. (2007) provides a more descriptive synopsis on land-surface interactions, Reick et al. (2013) complements both with an addition of land cover change processes, and Hagemann and Stacke (2015) introduces soil hydrological mechanisms within a

multilayer scheme applying five layers.

In JSBACH, the land surface is divided into grid-cells, which are split into bare soil and vegetative areas. The vegetative area is further divided into tiles representing the most prevalent vegetation classes, called plant functional types (PFTs) (Reick et al., 2013). In our site-level simulations, the model was set to use only one PFT, coniferous evergreen trees. The seasonal development of leaf area index (LAI) for the trees is regulated by air temperature and soil moisture with a single limiting value

(for all sites) for the maximum of LAI. This maximum value was fixed and the site-specific fractions of vegetative area were adjusted to reproduce the measured site level LAI.

The predictions of phenology are produced by the Logistic Growth Phenology (LoGro-P) sub-model in JSBACH (Böttcher et al., 2016). Photosynthesis is described by the biochemical photosynthesis model (Farquhar et al., 1980). Following Kattge et al. (2009), we set the maximum electron transport rate ($J_{max}$) at 25 degrees Celsius to 1.9 times the maximum carboxylation

rate ($V_{C,max}$), which is in line with e.g. Leuning (2002); Ueyama et al. (2016). The photosynthetic rate is dependent on the used stomatal conductance formulation, introduced in Section 2.3. Radiation absorption is estimated by a two stream approximation within a three-layer canopy (Sellers, 1985). Especially in sparse canopies, radiation absorption is affected by clumping of the leaves which is here taken into account according to the formulation by Knorr (1997).

Parameters detailing site-specific soil properties, such as soil porosity and field capacity, were derived from FLUXNET

datasets and the references in Table 1. We approximated the soil composition and generated these properties following Hagemann and Stacke (2015).

### 2.3  Modifications to the JSBACH model

All parameters of interest, presented in Table 2, were extracted from the JSBACH model code to an external file to facilitate the simulations. The default values of newly added parameters (not originally in JSBACH: $\tau$, $q$, $g_0$, $g_1$) were derived from a

synthesis of literature values. Most of the parameter ranges (limiting values for the parameters) were adapted from our previous work on a similar topic (Mäkelä et al., 2016). The parameter grouping was done to enhance optimisation and the mechanism is explained in Section 2.7. Group I consists of parameters most directly affecting photosynthesis, group II parameters are intimately involved with soil moisture, and group III are the logistic growth phenology (LoGro-P) model parameters. The equations governed by these parameters are presented in Appendix A.

**Table 2.** Descriptions of model parameters with default values, range of acceptable values and references to equations in the manuscript or in the appendices. Parameters in the same group were calibrated simultaneously.

| Parameter | def | range | Units | Group | Description | Eq. |
|---|---|---|---|---|---|---|
| $V_{C,max}$ | 62.5 | [40,65] | $\diamond$ | I | Farquhar model maximum carboxylation rate at 25°C of the enzyme Rubisco (coupled with maximum electron transport rate at 25°C with a factor of 1.9)   [$\diamond = \mu$ mol($CO_2$) m$^{-2}$ s$^{-1}$]. | A2 |
| $\alpha$ | 0.28 | [0.26,0.32] | - | I | Farquhar model efficiency for photon capture at 25°C. | A4 |
| $\tau$ | 10.0 | [5,15] | days | I | Adjustment period length in acclimation of photosynthesis. | 1 |
| $c_b$ | 5.0 | [4,7] | - | I | Multiplier in momentum and heat stability functions (Louis, 1979). | - |
| $f_{C3}$ | 0.87 | [0.7,0.95] | - | I | Ratio of unstressed C3-plant internal/external $CO_2$ concentration. | A3 |
| $q$ | 0.0 | [0,1] | - | I | Exponential scaling of water stress in reducing photosynthesis. | A1 |
| $g_0$ | 0.001 | [1E-5,5E-3] | $\nabla$ | I | Residual stomatal conductance   [$\nabla =$ mol m$^{-2}$ s$^{-1}$]. | B3 |
| $g_1$ | Values in Table 3 | | - | I | Slope of the stomatal conductance function. | B3 |
| $a$ | 2.8 | [1.5,3.5] | - | I | Base rate of stomatal conductance response to atmospheric humidity for the Friend and Kiang model. | B3 |
| $d$ | 80 | [50,120] | - | I | Exponential rate of stomatal conductance response to atmospheric humidity for the Friend and Kiang model. | B3 |
| $\theta_{dr}$ | 0.9 | [0.5,0.95] | - | II | Volumetric soil water content above which fast drainage occurs. | A6 |
| $\theta_{hum}$ | 0.5 | [0.2,0.8] | - | II | Fraction depicting relative surface humidity based on soil dryness. | A9 |
| $\theta_{pwp}$ | 0.35 | [0.15,0.4] | - | II | Volumetric soil moisture content at permanent wilting point. | 2 |
| $\theta_{tsp}$ | 0.75 | [0.25,0.8] | - | II | Value of volumetric soil moisture content above which transpiration is unaffected by soil moisture stress ($\beta$); and $0.9\theta_{tsp} \geq \theta_{pwp}$. | 2 |
| $p_{int}$ | 0.25 | [0.15,0.35] | - | II | Fraction of precipitation intercepted by the canopy. | A5 |
| $s_{sm}$ | 5.9E-3 | [1E-4,0.1] | m | II | Depth for correction of surface temperature for snow melt. | - |
| $w_{skin}$ | 2.0E-4 | [1E-5,5E-3] | m | II | Maximum water content of the skin reservoir of bare soil. | - |
| $C_{decay}$ | 13.0 | [5,25] | days | III | LoGro-P: memory loss parameter for chill days. | A12 |
| $S_{min}$ | 10.0 | [5,30] | °C days | III | LoGro-P: minimum value of critical heat sum. | A12 |
| $S_{range}$ | 150.0 | [100,300] | °C days | III | LoGro-P: maximal range of critical heat sum. | A12 |
| $T_{alt}$ | 4.0 | [2,10] | °C | III | LoGro-P: cutoff in alternating temperature. | A10 |
| $T_{ps}$ | 10.0 | [3,25] | °C | III | LoGro-P: memory loss parameter for pseudo soil temperature. | A14 |

The start of the growing season in the JSBACH model is defined by a "spring event" in the LoGro phenology model (appendix A3) that induces leaf growth. The phenology model calculates a sum of ambient temperature (heatsum) since last autumn that is above the cutoff value $T_{alt}$, presented in Eq. (A10). It also calculates a variable threshold, defined in (A12), for the heatsum to reach. The threshold decreases based on the number of days the ambient temperature is below $T_{alt}$, whereas the heatsum increases. When the heatsum reaches the threshold, the plant leaves are free to grow.

However, coniferous evergreen trees do not shed all of their leaves for winter and the existing foliage enables them to quickly initiate photosynthesis in the following spring. The start of the photosynthetically active season in the model has been observed to occur too early in the Boreal region by e.g Böttcher et al. (2016). In order to correct this behaviour i.e. to restrain the respiration and photosynthesis of conifers in the early spring, we utilise a delayed effect of temperature for photosynthetic activity, introduced by Mäkelä et al. (2004). To calculate the reduction, we must first define the state of photosynthetic acclimation that Mäkelä et al. (2004, p.371) present as: "an aggregated measure of the state of those physiological processes of the leaves that determine the current photosynthetic capacity at any moment".

The state of acclimation ($S$) is calculated from air temperature ($T$) with a delay prescribed by parameter $\tau$ (this is similar to the calculation of $T_S$ in appendix A14). $S$ is then inserted into sigmoidal relation Eq. (1) to calculate a factor $\gamma$, a formulation that is adapted here from Kolari et al. (2007). Finally, $\gamma$ is used to reduce the photosynthetic efficiency in Eq. (A1). $T_{1/2}$ denotes the inflection point where $\gamma$ reaches half of $\gamma_{max}$, $k$ is the curvature of the function and $\gamma = 1$ when $S \geq 10$.

$$\frac{dS}{dt} = \frac{T - S}{\tau}, \qquad \gamma = \frac{\gamma_{max}}{1 + e^{k(S - T_{1/2})}} \tag{1}$$

The JSBACH model was also modified to include altogether six different stomatal conductance formulations following Knauer et al. (2015). These formulations include the pre-existing Baseline and Bethy versions as well as the Ball-Berry model and three of its variants. Model information is gathered in Table 3 for easy referencing and the detailed formulations are given in appendix B. The limits of the slope of the stomatal conductance formulation parameter ($g_1$) were set to reflect commonly observed values from physiological measurements (Egea et al., 2011). The limits of $g_1^{USO}$ reflect the results presented by Lin et al. (2015).

**Table 3.** Stomatal conductance models with default values and range for $g_1$ and references to equations in Appendix B as well as related articles. The $\star$ symbol indicates the Ball-Berry model and its variants.

| Stomatal conductance model | short | $g_1$ | range | references | |
|---|---|---|---|---|---|
| Baseline | Base | - | - | B1 | Knorr (1997) |
| Biosphere-Energy-Transport-Hydrology | Bethy | - | - | B2 | Knorr (2000) |
| $\star$ Ball-Berry | BB | 9.0 | [4,10] | B3 | Ball et al. (1987) |
| $\star$ Leuning | Leu | 8.0 | [6,10] | B3 | Leuning (1995) |
| $\star$ Friend and Kiang | F&K | 9.5 | [7,11] | B3 | Friend and Kiang (2005) |
| $\star$ Unified stomatal optimisation | USO | 2.0 | [1.5,3.5] | B3 | Medlyn et al. (2011) |

We have also included two additional parameters ($a$ and $d$ in Table 2) for the Friend and Kiang (Friend and Kiang, 2005) stomatal conductance formulation in B3. These parameters were not originally included in the optimisation, but the resulting cost function (9) values were poor when compared to the other formulations. At that point, these parameters were included in the optimisation process. This increases the degrees of freedom for the Friend and Kiang model by two and therefore may give it an advantage when compared to the other Ball-Berry type formulations, which has to be considered in the interpretation of the results.

All of the stomatal conductance models contain an empirical water stress factor $\beta$, which reduces stomatal conductance as a function of volumetric soil water content ($\theta$).

$$\beta = \begin{cases} 1, & \theta \geq \theta_{tsp} \\ \frac{\theta - \theta_{pwp}}{\theta_{tsp} - \theta_{pwp}}, & \theta_{pwp} < \theta < \theta_{tsp} \\ 0, & \theta \leq \theta_{pwp} \end{cases} \tag{2}$$

In JSBACH, the stomatal conductance ($g_s$) is primarily resolved to estimate carbon fixation. The same $g_s$ is then later used
to calculate transpiration (A8). In the original JSBACH formulation (i.e. the Baseline version), the $g_s$ is first resolved for unstressed canopy and then scaled by the water stress factor $\beta$. The Bethy approach is similar, but the conductance can also be limited by water supply (B2). In cases when the water supply is not the limiting factor, the calculations are similar to the Baseline version. In all of the empirical Ball-Berry variants, the stomatal conductance can be written as $g_s = g_0 + c\beta g_1$. The residual conductance ($g_0$) and the slope of the function ($g_1$) are both formulation specific parameters as well as the factor $c$,
that incorporates net photosynthesis and effects of atmospheric humidity and $CO_2$ concentration. The parameters $g_0$ and $g_1$ are part of our sampling and optimisation processes (group I in Table 2 when applicable).

The water stress factor ($\beta$) limits the carbon fixation and transpiration via the stomatal conductance formulation. Following Egea et al. (2011), it is also used to directly limit the net assimilation rate ($A_n$), as seen in (A1). The additional scaling (or limiting) factor for $A_n$ takes the form $\beta^q$, so it is a function of both soil water content $\theta$ and the parameter $q$. Maximal reduction
is achieved when $q = 1$ and the reduction factor reverts to $\beta$. The minimal reduction occurs when $q = 0$ and the reduction factor resembles a step function (at $\theta = \theta_{pwp}$). For any other value of $q$, it is a continuous convex function between the two extremes $\beta^q : [\theta_{pwp}, \theta_{tsp}] \rightarrow [0, 1]$.

## 2.4 Model simulations

The site level measurements, used as model inputs, are air temperature, air pressure, precipitation, humidity, wind speed and
$CO_2$ concentration as well as short- and longwave and potential shortwave radiation. Additionally, evapotranspiration (ET) and gross primary production (GPP), derived from the eddy covariance (EC) measurements, are used to constrain and evaluate the model (as explained later in Sections 2.8 and 2.9). We drive the model with half-hourly data but output daily values.

The initial state of the JSBACH model can be generated from predefined values of state variables (usually empty initial storage pools) or the model can be restarted from a file describing the state of some previous run. Depending on the area of
interest, a model spin-up may be required to bring the model into a steady state. In our simulations, some of the more slowly changing variables (e.g. soil water content and LAI) need to be equilibrated, so a spin-up is required. This can be achieved by running the model over a set of measurements multiple times, each time restarting from the final state of the previous run.

The calibration period consists of the first five years given for the calibration sites in Table 1. The spin-up is achieved by looping over these five years, altogether four times (20-year spin-up) and then saving the state of the model at the end of the
run. The actual calibration is started from the beginning of the calibration period, using the previously saved state variables. To

reduce any bias this induces, the first year in the calibration run is removed from the cost function calculations. The spin-ups for the validation sites in Table 1 are similarly generated.

During the summer 2006, the Hyytiälä (FI-Hyy) measurement site suffered from a severe drought (Gao et al., 2017), leading to visible discolouration of needles. These events are difficult for models to capture and hence are of interest to modellers. We have previously and unsuccesfully attempted to optimise the JSBACH model (Mäkelä et al., 2016) for this event. Here we focus directly on the extended dry period (190–260th day of the year in 2006), during which the actual drought is mostly in effect between 210–235th DOY. We adjusted some of the parameter values as those uncovered by the more general calibration, presented above. The spin-up was the same as for the calibration period, but at the end of the spin-up, the model was run forward to the start of the year 2006. Only values between the 190–260th day of the year (DOY) in 2006 were used in constraining the model.

## 2.5 Sampling process

We describe the modelling setup with the equation $\mathbf{y} = \mathcal{M}(\boldsymbol{\theta}, \mathbf{x}) + \mathbf{e}$, where the aim is to reproduce the observations ($\mathbf{y}$) with our model ($\mathcal{M}$), the driving data ($\mathbf{x}$) and the current parameter values ($\boldsymbol{\theta}$). The residuals ($\mathbf{e}$) depict how well the model reproduces the observations and they form the basis of the likelihood function (formulated in Section 2.9), that is used to derive the parameter posterior distributions.

Using Bayes' rule on conditional probability we can write the parameter posterior density ($p(\boldsymbol{\theta}, \mathcal{M}|\mathbf{x})$) as a function of the likelihood ($\mathcal{L}(\mathbf{x}|\boldsymbol{\theta}, \mathcal{M})$), parameter prior distributions ($\pi(\boldsymbol{\theta})$) and the model evidence ($Z(\mathbf{x}|\mathcal{M})$). As usual and from here on, we do not write $\mathcal{M}$ in the Bayes' formula:

$$p(\boldsymbol{\theta}|\mathbf{x}) = \frac{\mathcal{L}(\mathbf{x}|\boldsymbol{\theta})\pi(\boldsymbol{\theta})}{Z(\mathbf{x})} \tag{3}$$

We can now utilise the posterior density as a probability density for the parameters and infer the expectation values:

$$E[\boldsymbol{\theta_i}] = \frac{1}{Z} \int \boldsymbol{\theta_i} p(\boldsymbol{\theta}|\mathbf{x}) d\boldsymbol{\theta}, \qquad Z = \int p(\boldsymbol{\theta}|\mathbf{x}) d\boldsymbol{\theta} \tag{4}$$

Above $\boldsymbol{\theta}_i$ is the $i$-th element of the parameter vector. Generally, Eq. (4) cannot be analytically solved, hence it is usually estimated numerically. Commonly this is achieved by one of the many Markov chain Monte Carlo (MCMC) methods, but in this study we apply the adaptive population importance sampler (APIS) defined by Martino et al. (2015). APIS (Martino et al., 2015) is a Monte Carlo (MC) method that utilises a population of importance samplers (IS) to jointly estimate the target pdf ($p(\boldsymbol{\theta}|\mathbf{x})$) and the normalising constant ($Z(\mathbf{x})$) by a deterministic mixture approach (Veach and Guibas, 1995; Owen and Yi, 2000), whereas the MCMC methods do not care about the value of $Z$. We denote the importance sampling density as $q(\boldsymbol{\theta})$.

$$E[\boldsymbol{\theta_i}] = \frac{1}{Z} \int \boldsymbol{\theta_i} r(\boldsymbol{\theta}) q(\boldsymbol{\theta}) d\boldsymbol{\theta}, \qquad \text{where} \qquad r(\boldsymbol{\theta}) = \frac{p(\boldsymbol{\theta}|\mathbf{x})}{q(\boldsymbol{\theta})} \tag{5}$$

Above $r$ is the reweighing factor that is the driving force in importance sampling. We will next give a summary description of the sampling process with comparison to a general multichain MCMC approach (since MCMC methods are more commonly used in these types of situations).

1. The initialisation of a multichain MCMC sampler and APIS are very similar. In our simulations, APIS is set up as 40 simultaneous and independent importance samplers. This is similar to an independent 40-chain MCMC sampler. Each sampler or chain has a random starting location drawn from a uniform distribution defined by the parameter ranges, given in Table 2. The initial sampling (or prior) distribution for each sampler is also randomly generated – we use truncated Gaussian distributions with diagonal covariance matrices, where the standard deviations are randomised. The sampling distributions will evolve throughout the process.

2. In an MCMC setup, the model would be run once (for each chain), evaluated and then the draw (parameter values) accepted or rejected accordingly. In APIS, instead of a single element (one run) we use a sample size of 50. This means that we draw 50 elements with each IS sampler (or "chain") independently. These draws are then evaluated and reweighted as presented in Eq. (5).

3. The 50 reweighted draws (for each IS sampler separtely) are used to calculate a new location for the sampling distribution. This location is automatically accepted (no rejection criteria) and we also adapt the shape of the distribution using the self-normalising AMIS estimator by Cornuet et al. (2012).

4. Additionally, all of the draws in APIS are used to calculate "global" estimates of the parameter expected values. This process utilises the deterministic mixture approach (Veach and Guibas, 1995; Owen and Yi, 2000) and is fully iterative with no need for any recalculations as the previous estimates are directly adjusted (no information is lost either).

MCMC chains track the evolution of single elements, and occasionally adjust the sampling distribution. The sample size in APIS is larger (it is not a Markov chain method) and the focus is on the evolution of the locations of the sampling distributions, not on the individually drawn elements. These location parameters are expected to be around all the modes of the target and the deterministic mixture ensures the stability of the estimation of the (global) parameter expected values. As an importance sampler, APIS is also a variance reducing method.

Before taking a more detailed look at APIS, we make some further notes about the sampling process. The first element of the 50 draws (item 2 in the list above) is always fixed as the current mean. We run the spin-up (Section 2.4) and generate the model starting state only for the proposal means, and use the same state for the other 49 draws (perturbed around the proposal mean). This requirement stems from a need to reduce computational time as running the model to a steady state is costly. This approach might induce some discrepancies, but they are mitigated by removing the first year of the calibration simulations (as explained in Section 2.4). We also slightly reduce the importance weights of the 49 samples (more reduction for samples further from the proposal mean), when calculating the new location parameters (item 3 in the list above) – the reduction only (slightly) slows the adaptation of the IS sampler locations. Finally, we note that this approach ensures that we run the proposal means, that are the focus in APIS, with the correct spin-up.

## 2.6 Adaptive population importance sampler

Normally, only the location parameters of the IS proposals are adapted, but we also adapt the shape parameters using the self-normalising AMIS estimators by Cornuet et al. (2012). APIS is able to utilise different or a mixture of normalised proposals densities, but we use truncated Gaussian proposals with diagonal covariance matrices.

In our simulations, APIS is formed of $40$ independent IS estimators. Each estimator draws a sample $\boldsymbol{\theta}_i, i \in \{1,...,N\}$, of size $N = 50$ at a time from their own proposal distribution $q_j(\boldsymbol{\theta}), j \in \{1,...,M\}, M = 40$. The estimator then calculates the importance weights ($w_{ij} = \frac{p(\boldsymbol{\theta}_i|\mathbf{x})}{q_j(\boldsymbol{\theta}_i)}$) for each sample. The location ($\boldsymbol{\mu}_j$) and shape ($\mathbf{C}_j$) parameters (Cornuet et al., 2012) of each proposal are updated using only samples (and weights) drawn from $q_j$. The new shape parameters are formed as a mean of the previous estimate and $\mathbf{C}_j$, as calculated below.

$$\boldsymbol{\mu}_j = \frac{\sum_i w_{ij} \boldsymbol{\theta}_i}{\sum_i w_{ij}}, \qquad \mathbf{C}_j = \frac{\sum_i w_{ij} (\boldsymbol{\theta}_i - \boldsymbol{\mu}_j)(\boldsymbol{\theta}_i - \boldsymbol{\mu}_j)^T}{\sum_i w_{ij}} \tag{6}$$

The simple IS estimators alone are rarely sufficient if the target is even slightly complicated. One classical way of tackling this problem is to join multiple IS estimators together. The simplest approach is to calculate the weights for each of these estimators separately and to normalise the result by the combined sum of all weights. However, this leaves the estimators susceptible to "bad" proposals. APIS suppresses the bad proposals by utilising the deterministic mixture approach (Veach and Guibas, 1995; Owen and Yi, 2000) presented in Eq. (7), where each proposal $q_j$ is evaluated at all the drawn samples and weighed by the amount of samples drawn ($N_j = 50$) from that proposal. This is equivalent to joining the normalised proposal densities together and evaluating the joint pdf.

$$w_{ij} = \frac{p(\boldsymbol{\theta}_{ij}|\mathbf{x})}{\sum_j \left(\frac{N_j}{\sum_k N_k}\right) q_j(\boldsymbol{\theta}_{ij})} \tag{7}$$

The parameter expectation values and the normalising constant in Eq. (5) can now be estimated by Monte Carlo integration using weights calculated in Eq. (7).

## 2.7 Parameter optimisation

The APIS algorithm is a rather robust method meant for examining the full target probability distribution and locating the modes of the target distribution. Adaptation in APIS utilises multiple draws simultaneously, which can easily lead to few parameters controlling this process (the marginal density of one or few parameters dominates the calculations). Since we also did not run the model spin-up for all drawn samples (although the discrepancies should be minimal), we utilise a simple custom stochastic optimiser to locate the optimal set of parameter values. This optimiser is run after the APIS calibration simulations and separately for the drought period. The optimiser utilises the exact same datasets (calibration, validation, observations etc.) as APIS, the spin-up is generated for all drawn samples separately and the initial state of the algorith is the mean value of the APIS final configuration (location parameters).

Our optimiser is a simple random sampler amplified by the "velocity" of the last jump (the idea is similar to Hamiltonian or Hybrid Monte Carlo by Duane et al. (1987)). We draw a set of samples from a small Gaussian proposal distribution in the

vicinity of the current best estimate and calculate the cost function for the samples. Whenever a better point is found (smaller cost function), we jump to that (update the mean of the proposal distribution). The "velocity" of the jump (for us merely distance of change in each parameter) is then added to the new mean (with a maximal limit of one standard deviation in the proposal distribution), but it is reduced and eventually removed if a better sample is not found.

The covariance matrix of the proposal distribution is recalculated at predefined intervals (for all parameters). Additionally, we utilise a subset sampling procedure, where the samples are first drawn from the full parameter space, in the next step they are drawn only from group I in Table 2 (the rest are kept at their current optimal values), followed by groups II and III and then back to the full parameter space. When the number of parameters is reduced, we are more likely to find a better set of parameter values. We have kept the parameters mostly affecting the same processes in the same group, but some dependencies
may not be apparent and hence it is also important to draw samples from the full parameter space.

## 2.8 Simulation analysis

Even though APIS is not a Markov chain method, we can (naively) interpret the evolution of the location paramaters of each IS sampler as chains. The resulting 40 chains have random starting positions but they are relatively short (we present results from the Bethy calibration, where the chains were adjusted 100 times), hence we did not discard any of the samples. We test
the convergence of these chains with the Gelman-Rubin diagnostic tests (Gelman and Rubin, 1992), comparing the variance between the chains to the variance within each chain, and calculating the potential scale reduction factors ($\hat{R}$). We also test the stability of the (parameter) global expected value estimate (using the deterministic mixture approach) by calculating the difference of the final global expected value and the mean of the location parameters (at each iteration). We denote this test as $\delta$ and report the number of the iterations when this difference is below 5% of the parameters range, given in Table 2.

In order to visualise the results, we have utilised a Gaussian kernel density estimation (KDE) to produce distributions from the APIS simulation location parameters. In practice, KDE places a Gaussian distribution centred at each sample and the constructed composite distribution is an estimate of the underlying actual distribution. The bandwidth for the distributions is calculated using the Scott's rule (Scott, 2004): the data covariance matrix is multiplied by a factor $n^{\frac{-1}{d+4}}$, where $n$ is the number of data points and $d$ is the number of dimensions.

The effectiveness of each parameter was calculated from the final state of each optimisation process. This was done by first setting all parameters to their optimised values. Then we (evenly) sampled each parameter separately from their range of acceptable values, given in Table 2, and calculated the corresponding cost functions. For each parameter the maximum difference in these cost function values (and the optimised value) was recorded. The parameters (within each optimisation) were then ordered by these numbers (with highest difference meaning highest effectiveness) and separated into three groups
with highest (most effective) and lowest (least effective) effectiveness values, and the rest. This effectiveness relates to how the APIS "sees" the sampling process – the 50 draws are evaluated simultaneously and a very effective parameter can easily mask the influence of a less effective (the marginal density of one or few parameters dominates the calculations).

  We report the slope of the regression line ($b$) and the coefficient of determination ($r^2$), between the observations ($y_i$) and the model output ($x_i$). The slope of the regression line is highly indicative of the model bias (difference of the expected values of

the observations and the model). Hence we interpret the bias directly from $b$ (in our results the regression lines pass near origin so the differences this induces are negligible).

$$b = \frac{\sum_i (x_i - \overline{x_i})(y_i - \overline{y_i})}{\sum_i (y_i - \overline{y_i})^2}, \qquad r^2 = 1 - \frac{\sum_i (x_i - y_i)^2}{\sum_i (y_i - \overline{y_i})^2} \tag{8}$$

## 2.9 Cost function

The Bayesian framework requires a likelihood function that optimally combines pointwise model and observational errors. The JSBACH model error is unknown as is the (pointwise) observation error. We could use a general type of error estimate (such as that of 20% of the flux value) for the observations, but would have to include a minimal site and instrumentation dependent precision. In this study, the full error is treated as Gaussian white noise. Because of these limitations, we are calling and defining our likelihood as a cost function. It is calculated with the same parameter values for each site, using site spesific daily
measurements with the gap-filled, low-quality and winter (between the 315th and the 75th day of the year) values removed (resulting in $N_{ET}$ and $N_{GPP}$ points). These site level estimates are averaged to produce the actual cost function, which is then returned for the algorithm to produce an estimate that is independent of the characteristics of any single site.

The cost function (9) in our simulations is based on the normalised mean squared error (NMSE) estimates of the daily gross primary production (GPP) and the daily evapotranspiration (ET). The residual of each variable is divided by the mean
of observations, as has been previously done by e.g. Mäkelä et al. (2016); Knauer et al. (2015); Groenendijk et al. (2010); Trudinger et al. (2007). We make use of this approach since we needed to balance two series of different magnitudes (ET and GPP). The residuals are additionally divided by the (site specific) number of observations so that the cost function is not biased towards any specific site. The cost function (without the normalisation) can be interpreted as a negative log-likelihood function with a (Gaussian) error term equal to the observational mean.

$$cf_1 = \overbrace{\frac{1}{N_{ET}} \sum \left( \frac{ET_{mod} - ET_{obs}}{\overline{ET_{obs}}} \right)^2}^{\text{NMSE}_{ET}} + \overbrace{\frac{1}{N_{GPP}} \sum \left( \frac{GPP_{mod} - GPP_{obs}}{\overline{GPP_{obs}}} \right)^2}^{\text{NMSE}_{GPP}} \tag{9}$$

We also use a modified version of this cost function, where the NMSE's are weighted by factors based on coefficients of determination ($r^2$) defined in Eq. (8). This latter cost function is only used during the separate drought period optimisation for Hyytiälä. During the drought we are more interested in the correct timing of the change in GPP and ET fluxes, rather than the size of the actual change. The aim is to correctly reproduce the changes in the water use efficiency (WUE) of plants, which we
interpret here as the pointwise ratio of (ecosystem level) GPP to ET. The NMSE values ensure that the overall amplitude of the fluxes will remain satisfactory.

$$cf_2 = (1 - r_{ET}^2)\text{NMSE}_{ET} + (1 - r_{GPP}^2)\text{NMSE}_{GPP} \tag{10}$$

## 3   Results

First we present the performance of the APIS algorithm and the parameters themselves, followed by site and stomatal conductance model specific results, and finally an examination of the Hyytiälä drought event in 2006. For simplicity, we use the name of the stomatal conductance model to refer to the JSBACH model utilising that stomatal conductance formulation.

The evolution of the APIS algorithmic process is presented in Fig. 1 for three parameters from the calibration of the Bethy model. The chosen parameters highlight different levels of identifiability for the algorithm (with the given cost function). The first parameter ($f_{C3}$) shows a well identifiable situation, where the algorithm quickly locates the area of high probability. The second parameter ($\theta_{dr}$) is also identifiable but the speed of convergence is diminished. The last example ($C_{decay}$) represents situations where the parameter is not constrained. We have included images of the APIS chains for the other parameters as

supplement S1 along with parameter posterior estimates at 20 iterations with the Bethy and Ball-Berry formulations.

We also report the results of the Gelman-Rubin (Gelman and Rubin, 1992) and $\delta$ tests in Table 4. Both of these tests indicate that the algorithm is performing well at 20 iterations – the values of $\hat{R} \approx 1$, which means that further simulations are unlikely to improve the variance estimates. However, for some parameters, the convergence of the global estimate is slow (as also seen in the supplementary image S1 for e.g. $\tau$, $c_b$ and $q$). The APIS sampling process did not reveal any multimodal distributions

and thus provided suitable initial conditions for the optimisation.

**Table 4.** Parameter scale reduction $\hat{R}$ (at APIS iteration) and stability $\delta$ (threshold number of iterations) estimates from the Bethy simulations.

|  | $V_{C,max}$ | $\alpha$ | $\tau$ | $c_b$ | $f_{C3}$ | $q$ | $\theta_{dr}$ | $\theta_{hum}$ | $\theta_{pwp}$ |
|---|---|---|---|---|---|---|---|---|---|
| $\hat{R}$ at 20 | 1.12 | 0.99 | 1.02 | 0.99 | 1.0 | 0.99 | 1.0 | 1.3 | 1.08 |
| $\hat{R}$ at 100 | 1.3 | 1.03 | 1.25 | 1.16 | 1.03 | 1.08 | 1.03 | 1.52 | 1.16 |
| $\delta$ ($\pm 0.05$) | 20 | 21 | 27 | 40 | 0 | 36 | 18 | 14 | 17 |
|  | $\theta_{tsp}$ | $p_{int}$ | $s_m$ | $w_{skin}$ | $C_{decay}$ | $S_{min}$ | $S_{range}$ | $T_{alt}$ | $T_{ps}$ |
| $\hat{R}$ at 20 | 0.99 | 1.01 | 0.99 | 1.0 | 0.99 | 0.99 | 0.99 | 0.99 | 0.99 |
| $\hat{R}$ at 100 | 1.06 | 1.13 | 1.0 | 0.99 | 0.99 | 1.0 | 0.99 | 0.99 | 0.99 |
| $\delta$ ($\pm 0.05$) | 26 | 35 | 8 | 0 | 12 | 22 | 0 | 1 | 0 |

### 3.1   Optimised parameters

The results of the optimisation process are gathered in Table 5. There is an overall agreement on the values of the most prevalent parameters (see the bold and italic characters in Table 5 between the models). Most notably, the permanent wilting point ($\theta_{pwp}$) and the point above which transpiration is unaffected by soil moisture stress ($\theta_{tsp}$) have been significantly lowered. The LoGro

phenology parameters, which affect the timing of the spring and autumn events, are expected to contribute only little to the cost function. The coniferous evergeen trees do not shed all their leaves for winter and therefore the timing of the bud burst is not as critical as for e.g. deciduous trees. Additionally, because of the existing foliage, the state of acclimation parameter $\tau$ that

depicts the reduction of carbon assimimilation in the early spring likely dominates the phenology parameters that determine when new leaves start to grow.

**Table 5.** Parameter default and optimised values for the calibration period with corresponding cost function value. The values written in boldface were the most effective and the italic values the least effective for the given experiment. Also presented are the fixed parameter values for the drought period optimisation, with opt referring to the use of the corresponding optimised value from this table.

| Parameter | def | Base | Bethy | BB | Leu | F&K | USO | dry set |
|---|---|---|---|---|---|---|---|---|
| $V_{C,max}$ | 62.5 | 48.4 | 57.1 | 55.4 | 49.7 | 50.8 | **50.5** | 52.0 |
| $\alpha$ | 0.28 | 0.318 | 0.318 | 0.319 | 0.317 | 0.319 | 0.318 | 0.318 |
| $\tau$ | 10.0 | 14.6 | 15.0 | 14.8 | 14.9 | 14.7 | 14.8 | 14.8 |
| $c_b$ | 5.0 | 5.4 | 4.1 | 6.7 | *4.4* | *4.3* | 4.6 | 5.0 |
| $f_{C3}$ | 0.87 | **0.75** | **0.83** | - | - | - | - | Table 7 |
| $q$ | 0.0 | *0.03* | 0.94 | *0.62* | *0.60* | *0.82* | *0.65* | Table 7 |
| $g_0$ | 1.0E-3 | - | - | 4.7E-3 | 4.7E-3 | 4.4E-3 | 4.2E-3 | Table 7 |
| $g_1$ | Table 3 | - | - | **9.9** | **8.8** | 10.9 | 1.6 | Table 7 |
| $a$ | 2.8 | - | - | - | - | 3.2 | - | opt |
| $d$ | 80 | - | - | - | - | **71** | - | opt |
| $\theta_{dr}$ | 0.9 | 0.86 | *0.65* | 0.88 | 0.83 | 0.8 | *0.90* | 0.85 |
| $\theta_{hum}$ | 0.5 | **0.2** | **0.2** | **0.21** | **0.2** | **0.2** | **0.2** | Table 7 |
| $\theta_{pwp}$ | 0.35 | **0.16** | **0.15** | **0.17** | **0.15** | **0.16** | **0.15** | Table 7 |
| $\theta_{tsp}$ | 0.75 | **0.31** | **0.35** | **0.3** | **0.31** | **0.32** | **0.33** | Table 7 |
| $p_{int}$ | 0.25 | 0.35 | 0.35 | 0.35 | 0.35 | 0.35 | 0.35 | 0.35 |
| $s_m$ | 5.9E-3 | 0.099 | *0.094* | *0.097* | *0.098* | 0.097 | *0.078* | 0.097 |
| $w_{skin}$ | 2.0E-4 | **3.7E-4** | **3.1E-4** | **3.5E-4** | **3.6E-4** | **3.3E-4** | **3.2E-4** | 3.4E-4 |
| $C_{decay}$ | 13.0 | *17.0* | *22.2* | *23.3* | *23.3* | *24.9* | *13.9* | opt |
| $S_{min}$ | 10.0 | *29.2* | *26.3* | *10.7* | 6.3 | *26.1* | 6.3 | opt |
| $S_{range}$ | 150 | *247* | *176* | *162* | *157* | *202* | *223* | opt |
| $T_{alt}$ | 4.0 | 2.0 | 2.8 | 5.8 | 8.2 | 2.5 | 8.3 | opt |
| $T_{ps}$ | 10.0 | *18.6* | 24.4 | 3.8 | 3.2 | 15.0 | 3.1 | opt |
| $cf_1$ | | 0.571 | 0.531 | 0.521 | 0.529 | 0.518 | 0.528 | |

Some of the parameters have converged to their limiting values, which can reflect deficiencies in the model structure or the preset parameter ranges. Convergence to the boundary can also be a problem in model calibration, but in this experiment, the algorithms were able to cope with the situation as APIS located the area of high probability and the optimiser located the maxima. The different parameter effectiveness levels reported in Table 5 can be roughly equated to the identifiability situations in Fig. 1. The effectiveness levels are highly situational (e.g. they depend on the sampling limits in Table 2 given for each

parameter) and merely reflect the parameter identifiability in the APIS process. Low effectiveness complements the test results in Table 4, as the tests may indicate good performance for a parameter (e.g. for $S_{range}$) that is ineffective in the simulations.

## 3.2 Annual cycles

We present the average annual cycles for the validation period and for all sites in Fig. 2 using the Bethy formulation that is
part of the standard model. The annual cycles generated with the other stomatal conductance models are added as supplement S2. The parameters of the regression lines ($b$ and $r^2$) between the measured and modelled ET and GPP fluxes of all the models are gathered in Table 6. These indicators have been calculated using all corresponding values regardless of the quality of the data. The sites are in the same order as in Table 1 with the six calibration sites first, followed by the four sites used only for validation. We have also included a supporting synthesis of the $b$ and $r^2$ values between the model simulations with the default
and optimised parameter values as supplement S3.

The optimisation has improved the model bias and the correlation coefficients for the GPP in Fig. 2 for nearly every site, with the exception of deteriorating bias for Poker Flat (US-Prr) and Zotino (RU-Zot). Additionally, the improvement in the timing of the springtime increase in the GPP is apparent. All of the correlation coefficients for the ET in Fig. 2 have also been improved but the model bias has mostly increased.

## 15 3.3 Drought event

The resulting parameter values, from the optimisation during the drought conditions in Hyytiälä (FI-Hyy) in the summer of 2006, are presented in Table 7. Setting the maximum carboxylation rate to a constant value ($V_{C,max} = 52.0$) enabled the full use of the dynamical range of $q$ – the idea was to ensure that $V_{C,max}$ does not dominate the optimisation, any value for $q$ is possible and it is able to influence the outcome. The LoGro phenology parameters and $\tau$ were fixed to their optimised values,
presented in Table 5, as they should not be affected by the drought. Likewise, the values of other parameters (not presented in Table 7) were set as compromises between the stomatal conductance formulations.

We can now compare the parameter values in Table 7 to those in Table 5. The values of the relative humidity parameter ($\theta_{hum}$) and the residual stomatal conductance ($g_0$) have remained nearly unchanged, but for the rest of the parameters have quite varied values. The leaf internal-to-external $CO_2$ concentration ($f_{C3}$) as well as the slope of the stomatal conductance ($g_1$)
are at the lower bound (expect $g_1$ for BB). Noticeably, the USO optimisation only changes the values of $\theta_{tsp}$ and $q$, and leaves the rest of the parameters almost untouched.

The changes these different parameterisations have on the model output are visualised in Fig. 3. All of the stomatal conductance models, with default parameterisation, suffer from too low ET values before (and during) the actual drought. This behaviour was corrected during the general optimisation, but has partially re-emerged with the dry period parameters for the
Baseline, Ball-Berry, Leuning, and to a lesser degree the Friend and Kiang formulations. Most of the models also exhibit too high ET values during the actual drought with the generally optimised parameter values. This behaviour was also corrected with the dry period optimisation, but the Baseline and especially the Bethy model now suffer from a too strong drawdown of ET. These models also demonstrate the too strong drawdown for the GPP. The GPP itself was greatly improved with both

**Table 6.** Slope of the regression line ($b$) and the coefficient of determination ($r^2$) for the different stomatal conductance formulations during the validation period with the optimised parameters. We have written the best values of $b$ and $r^2$ in boldface for each site, and italicised the abbreviations of the separate validation sites.

| | Evapotranspiration (ET) | | | | | | | | | | | |
| | $b$ | | | | | | $r^2$ | | | | | |
| Site | Base | Bethy | BB | Leu | F&K | USO | Base | Bethy | BB | Leu | F&K | USO |
|---|---|---|---|---|---|---|---|---|---|---|---|---|
| CA-Obs | 0.91 | 0.9 | **0.91** | 0.86 | 0.81 | 0.76 | 0.75 | **0.77** | 0.76 | 0.76 | 0.75 | 0.74 |
| CA-Qfo | 0.96 | 0.98 | **0.99** | 0.92 | 0.89 | 0.83 | 0.71 | **0.72** | 0.7 | 0.71 | 0.7 | 0.69 |
| FI-Hyy | 0.97 | 1.05 | 1.07 | 0.95 | **0.98** | 0.79 | 0.73 | **0.77** | 0.77 | 0.75 | 0.77 | 0.69 |
| FI-Ken | 0.54 | **0.64** | 0.62 | 0.56 | 0.58 | 0.48 | 0.48 | 0.51 | **0.52** | 0.49 | 0.51 | 0.45 |
| FI-Sod | 0.64 | 0.73 | **0.74** | 0.63 | 0.64 | 0.56 | 0.58 | **0.64** | 0.61 | 0.6 | 0.62 | 0.55 |
| RU-Fyo | 0.98 | 1.02 | 1.01 | 0.98 | **0.99** | 0.85 | 0.7 | 0.71 | 0.71 | 0.71 | **0.71** | 0.7 |
| *CA-Ojp* | 0.8 | **0.84** | 0.84 | 0.75 | 0.72 | 0.67 | 0.64 | **0.65** | 0.64 | 0.65 | 0.64 | 0.63 |
| *FI-Let* | 1.09 | 0.98 | 1.08 | 1.04 | **1.01** | 0.94 | 0.49 | 0.47 | 0.49 | 0.5 | **0.51** | 0.48 |
| *RU-Zot* | 0.49 | 0.56 | **0.56** | 0.47 | 0.46 | 0.41 | 0.45 | **0.52** | 0.5 | 0.47 | 0.48 | 0.41 |
| *US-Prr* | 0.38 | 0.37 | **0.42** | 0.35 | 0.33 | 0.35 | 0.48 | 0.53 | **0.53** | 0.46 | 0.44 | 0.43 |
| best values | 0 | 2 | 5 | 0 | 3 | 0 | 0 | 6 | 2 | 0 | 2 | 0 |

| | Gross primary production (GPP) | | | | | | | | | | | |
| | $b$ | | | | | | $r^2$ | | | | | |
| Site | Base | Bethy | BB | Leu | F&K | USO | Base | Bethy | BB | Leu | F&K | USO |
|---|---|---|---|---|---|---|---|---|---|---|---|---|
| CA-Obs | **0.83** | 0.77 | 0.82 | 0.81 | 0.81 | 0.77 | 0.87 | 0.9 | 0.89 | 0.89 | **0.91** | 0.9 |
| CA-Qfo | 0.97 | 0.95 | **0.98** | 0.96 | 0.96 | 0.9 | 0.84 | 0.87 | 0.85 | 0.86 | **0.88** | 0.87 |
| FI-Hyy | 1.02 | **1.01** | 1.05 | 1.03 | 1.06 | 0.98 | 0.94 | 0.94 | 0.94 | 0.95 | **0.95** | 0.95 |
| FI-Ken | 0.9 | 0.97 | **0.97** | 0.93 | 0.95 | 0.9 | 0.93 | 0.9 | 0.9 | 0.93 | 0.93 | **0.94** |
| FI-Sod | 0.66 | 0.71 | **0.71** | 0.67 | 0.69 | 0.65 | 0.88 | 0.87 | 0.86 | 0.89 | **0.9** | 0.9 |
| RU-Fyo | 0.95 | 0.88 | 0.91 | 0.96 | **0.98** | 0.91 | 0.89 | 0.88 | 0.88 | 0.91 | 0.91 | **0.91** |
| *CA-Ojp* | 0.72 | 0.74 | **0.75** | 0.7 | 0.69 | 0.66 | 0.83 | 0.85 | 0.84 | 0.85 | 0.86 | **0.86** |
| *FI-Let* | 1.27 | **0.99** | 1.09 | 1.25 | 1.26 | 1.21 | 0.93 | 0.88 | 0.89 | 0.94 | 0.94 | **0.94** |
| *RU-Zot* | 0.42 | **0.44** | 0.44 | 0.42 | 0.42 | 0.4 | 0.86 | 0.85 | 0.84 | 0.88 | **0.88** | 0.88 |
| *US-Prr* | 0.2 | **0.21** | 0.21 | 0.2 | 0.19 | 0.19 | 0.62 | 0.6 | 0.6 | 0.62 | **0.63** | 0.62 |
| best values | 1 | 4 | 4 | 0 | 1 | 0 | 0 | 0 | 0 | 0 | 6 | 4 |

optimisations and for all models. The dry period optimisation of the USO model also managed to correct the erroneous GPP of the general optimisation during the actual drought, where as the GPP of other formulations has remained roughly the same as with the general optimisation. The USO formulation results in the best fits for $r^2$ and $b$ with the dry period optimisation.

**Table 7.** Optimised parameter and corresponding cost function values with different stomatal conductance formulations for the extended dry period.

| Parameter | def | Base | Bethy | BB | Leu | F&K | USO |
|---|---|---|---|---|---|---|---|
| $f_{C3}$ | 0.87 | 0.7 | 0.7 | - | - | - | - |
| $q$ | 0.0 | 0.09 | 0.0 | 0.15 | 0.57 | 0.16 | 0.30 |
| $\theta_{tsp}$ | 0.75 | 0.57 | 0.46 | 0.48 | 0.44 | 0.45 | 0.41 |
| $\theta_{pwp}$ | 0.35 | 0.40 | 0.38 | 0.27 | 0.23 | 0.28 | 0.16 |
| $\theta_{hum}$ | 0.5 | 0.2 | 0.2 | 0.2 | 0.2 | 0.2 | 0.2 |
| $g_0$ | Table 3 | - | - | 4.9E-3 | 5.0E-3 | 3.8E-3 | 4.6E-3 |
| $g_1$ | Table 3 | - | - | 7.5 | 6.0 | 7.0 | 1.5 |
| $cf_2$ | | 0.42 | 0.44 | 0.39 | 0.41 | 0.41 | 0.41 |

The Bethy and the USO models demonstrate the most variability in the $\beta$-function values in Fig. 3 (rightmost panels), for the dry period optimisation. We selected these two stomatal conductance formulations to examine the changes to the water use efficiency (WUE) of plants during the extended dry period. The highlighted observations in Fig. 4 (rightmost panels) show a clear path of development for the drought where the observations imitate the letter $\delta$. The colourings follow the $\beta$-function values in Fig. 3 between the red vertical lines. Both observational colourings (same as the model colouring) are similar and depict, initially, a linear decrease in both ET and GPP, followed by a rapid decline in ET and a delayed decline in GPP. The recovery of plants from the drought can also be seen as the colouring starts to turn lighter. The models depict a more linear response of GPP to ET as the drought develops, although with USO we can see a bit more similarities in the pattern of the values.

Finally, we used both optimised parameter sets (Table 5 and 7) to produce the ET and GPP cycles for all sites and stomatal conductance models. This analysis (not shown) verified that in general conditions the Table 5 parameter values produced better estimates in general. The $b$ and $r^2$ values for the ET were systematically better for all stomatal conductance formulations (except one). There was some variation in the indicators for the GPP, where approximately a third of the values (of mostly $r^2$) are better with the dry period parameter set. These differences are mostly attributed to increased model bias (decreased $b$) that is explained by the lower values of $g_1$. Overall, the more general optimisation provided systematically better or comparable results to the dry period optimisation. The exception is the USO formulation, which had an approximately 1:1 distribution of best values for both variables in-between the parameter sets.

## 4 Discussion

We will first discuss the validity of our approach and the simulation setup, followed by examination of the success of the modifications made to the model, and close with some further remarks on the parameter values.

## 4.1 Validity of the simulations

Before we calibrated the model, we fixed the limiting value for LAI and adjusted the site-specific vegetative area fractions to reproduce the measured site level maximum of LAI. In the simulations, we focused on boreal coniferous forests, where light penetration is deep and the light conditions are homogenous – consequently we could assume a homogenous leaf distribution. Furthermore, the JSBACH model takes into account leaf clumping and we can assume the leaf orientation and shape to be similar throughout the study sites. Therefore, we argue that reproducing the site level maximum of LAI is appropriate approach in this study. Together with parameter calibration it has resulted in improved ET and GPP fluxes as can be verified from the $b$ and $r^2$ values in Fig. 2. The improvements in $b$ and $r^2$ are mostly seen in the GPP flux, which can be explained by the fact that the stomatal conductance in JSBACH is primarily resolved for carbon assimilation, and the same conductance is then used for transpiration (A8). Additionally, GPP is derived from the EC measurements by flux partitioning – this tends to remove some of the flux instabilities (that are still present in the ET).

We encountered difficulties in reproducing the fluxes for the validation sites with low LAI (i.e RU-Zot and US-Prr). This can be a consequence of the area scaling as the adjustment linearly changes the proportions between vegetative area and bare soil. Another reason is the lack of the site understory in these simulations. For example, approximately half of the $CO_2$ fluxes (and consequently roughly half of the GPP) for Poker Flat are produced by the site understory (Ikawa et al., 2015). Additionally, there are also many parameters describing site-specific soil properties (such as porosity) that were not part of the optimisation and may be inaccurate. These effects may also be pronounced due to the changes in parameters affecting soil moisture as well as the area scaling.

There were no clear differences between sites dominated by pine or spruce. Neither did we notice any particular effect on the bias, NMSE or correlation coefficient that could be explained by geographical location, stand age or annual precipitation or temperature. We optimised the model for individual (calibration) sites as well (not shown). Mostly this changed the values of parameters (such as $V_{C,max}$ and $g_1$) affecting the amplitude of the modelled fluxes. These parameters can be viewed to be more site-specific, a characteristics that is reduced in a multi-site calibration – the possibility of highly site-specific properties (and parameter values) can also explain the difficulties in reproducing the validation site observations. We are omitting these results as single-site optimisation can be viewed as overfitting the model and the results do not provide any additional insights.

The APIS performance tests (Gelman-Rubin and $\delta$) indicate that the algorithm is performing well at 20 iterations but the convergence of the global estimate for some parameters is slow. This is mostly a direct result of the normalisation of the cost function that inflates the target distribution, which reduces the parameter sensitivity to observations and gives too much weight to the initial locations and draws. Without the normalisation, the algorithm would also converge faster. Additionally, APIS is meant to examine the full target distribution with only some sequantiality – 20 iterations (or less) should be sufficient for APIS to locate the modes of the target. In longer APIS simulations, the global estimate would likely benefit from discarding the first half of the samples but this would require the estimate to be recalculated at each iteration (from the drawn samples) as it could not be calculated iteratively.

## 4.2 Delayed effect of temperature

We modified the JSBACH model by introducing the delayed effect of temperature for photosynthesis to restrain the respiration and photosynthesis of conifers in spring. The effect of this (delayed increase in GPP) is apparent in the annual GPP cycles of CA-Qfo, FI-Hyy, FI-Ken, FI-Sod and RU-Zot in Fig. 2. The delay is in place for the other sites as well, but the effect is less

apparent in the figure. This delay is to a lesser extent also reflected in transpiration, and consequently in ET, as can be seen e.g. at FI-Hyy and FI-Sod – for other sites this effect is not clear. The correction in the ET values can lead to an increase in model bias as is the case with Sodankylä (FI-Sod), where the too low autumn values in the default model were previously compensated by too high springtime values (in the sense of annual ET). Correcting the springtime behaviour leads to an increase in bias, but this should not be viewed as a fault in the optimisation as the model was previously mitigating an erroneous behaviour (too

low autumn ET) with another (too high springtime ET).

Mäkelä et al. (2004) used a linear dependency of photosynthetic efficiency to the state of acclimation, and reported 13.75 days to be the best fit for the adjustment period length ($\tau$). Kolari et al. (2007) utilised a sigmoidal relation and reported the value of 8 days, but noted that the range of values resulting in a good fit was large (5–10.4 days). Linkosalo et al. (2014) came to a similar conclusion when they encounter a near-flat distribution for $\tau$ in the range of 1–12 days. In our simulations $\tau$

exhibits larger optimal values (nearly 15 days), which is most likely due to the model adapting to the multi-site calibration (as sites have different characteristics, a longer acclimation period accounts better for these variations).

## 4.3 Stomatal conductance models

We examined the model behaviour with six stomatal conductance formulations and the resulting $b$ and $r^2$ values are presented in Table 6. The best performance (bolded values) in simulated ET is achieved by the BB model for bias and the Bethy formu-

lation for $r^2$. These two models also share the best performance in the GPP bias, whereas the best $r^2$ values for the GPP are demonstrated by the $F\&K$ model, followed by the USO formulation. Calculating the number of best values demonstrated by each model, we obtain that the best performance is shared by the Bethy (12) and F&K (12) formulations, followed by the BB (11) model. However, we note that some of the "best values" are only marginally better than comparable values. Additionally, we used two more parameters ($a$ and $d$) for the F&K formulation than for the other Ball-Berry formulations. Likewise, we

could have, for example, included the factor $D_0$ (B3) in the optimisation, which would have likely improved the performance of the Leuning model. Similarly to the results by Knauer et al. (2015), based on this (general) calibration, there is no clear single candidate for the best stomatal conductance formulation.

The model behaviour was also examined during the Hyytiälä drought of 2006. Some of the parameter values were kept fixed during these simulations, most of the fixed parameters should not affect the drought period calibration but there are exceptions,

such as the maximum carboxylation rate $V_{C,max}$. It can be argued that e.g. both the parameters $V_{C,max}$ and $g_1$ should decrease (Egea et al., 2011) during the drought but we decided to fix $V_{C,max}$ to get a better response for $q$. The best fit to the observations was achieved by the USO formulation, as seen in Fig. **??**, with remarkably similar parameter values to the general optimisation. The USO model was also able to (somewhat) replicate the "$\delta$" shape of the drought in Fig. 4.

The stomatal conductance function ($g_s = g_0 + c\beta g_1$) incorporates also the soil water parameters $\theta_{tsp}$ and $\theta_{pwp}$ in the form of the $\beta$-function as portrayed in Eq. (2). The changes in the values of these parameters (mostly $g_1, \theta_{tsp}$ and $\theta_{pwp}$) are intertwined. During the drought, the decrease in the optimised values of $g_1$ is expected as the plants close their stomata to minimise the loss of water by transpiration (Egea et al., 2011; Zhou et al., 2013). The same effect is also achieved by increasing the values of

$\theta_{tsp}$ and $\theta_{pwp}$ as this decreases the values of the $\beta$-function. The higher values of $g_1$ during the more general optimisation are better reflected by Franks et al. (2018), whereas the lower values during the drought are more in accordance with physiological observations by Egea et al. (2011). Likewise, Lin et al. (2015) found higher values for $g_1$ (both boreal area and gymnosperm trees) using the USO model.

In general, the site level estimates of ($g_0$ and) $g_1$ are sensitive not only to the stomatal conductance formulation but also

e.g. to the structure of the underlying model and the value of other parameters, such as maximum carboxylation rate ($V_{C,max}$). Wang (1996) reported $g_1 = 3.78$ (in Table 1, Control), using a Leuning model similar to ours, where $(1 + D_S/D_0)$ is replaced by $D_S$. Thum et al. (2007) approximated $g_1^{BB}$ to be 5 for Sodankylä while estimating the variation in the values of $V_{C,max}$ and maximum rate of electron transport $J_{max}$. We would suggest that the limiting values $\theta_{pwp}$ and $\theta_{tsp}$ should be optimised or fixed before introducing additional tuning factors such as mesophyll conductance or scaling the $\beta$ in multiple ways in the

stomatal conductance formulations (Egea et al., 2011). Our simulation setup for $q$ corresponds to the configuration 5 (C5) by Egea et al. (2011), with variables $q = q_B$ and fixed value $q_S = 1$.

## 4.4 Parameter values

Some of the parameters in this study have been calibrated before by e.g. Kattge et al. (2009); Knorr et al. (2010). Our approach differs from these as we required the model to reproduce the site level maximum of LAI. In contrast e.g. Knorr et al. (2010)

found the structural limit for (all-sided) LAI to be 4.2, which is considerably lower than the measured LAI for many of the sites in Table 1. Our approach directly scales the vegetative area, so it also scales GPP and also the amount of rain available for plants (as rain is directed to bare soil and vegetative area). This means that the parameter values should not be directly compared without taking the different paradigms into account. However, our optimised $V_{C,max}$ values are in-between 62.5 reported by Kattge et al. (2009) and 29.3 by Knorr et al. (2010), and are in line with the yearly cycle presented by Ueyama

et al. (2016).

The exponential scaling factor $q$ in Eq. (A1) of the $\beta$-function (2), was revealed to be ineffective in our optimisation as indicated in Table 5. In our simulations, this situation arises as the effective range of the $\beta$-function has been lowered by reducing $\theta_{pwp}$ and $\theta_{tsp}$. The actual soil moisture is rarely below the fraction $\theta_{tsp}$, so $q$ is constrained with a very limited number of datapoints, and thus has only minimal effect on the fluxes and the cost function. Therefore, the values presented for

$q$ in Table 5 can be unreliable and even unrealistic. This situation is remedied in the drought period optimisation when the soil moisture is low. The resulting values for $q$ in Table 7 have a wide dispersion, although they are mostly on the lower end. This signifies that the additional GPP reduction is mostly gradual, with a steep decrese near the permanent wilting point $\theta_{pwp}$.

The values of soil water parameters are closely grouped in the optimisations except for the values of $\theta_{pwp}$ during the drought. This can occur due to a larger impact, of the different stomatal conductance formulations on the accumulating soil

water content, than assumed – this can also be seen from the differences in the $\beta$-function values in Fig. 3. Furthermore, the values of $\theta_{tsp}$ and $\theta_{pwp}$ have been considerably lowered from their default values in both optimisations. This change can be perceived in at least two different ways. Either the boreal forests are not generally limited by soil moisture stress (except in the case of extreme drought) or the water retention capabilities of the soil (in the model) have been systematically overestimated.

The latter seems unlikely, in the light of results by e.g. Gao et al. (2016).

## 5   Conclusions

The adaptive population importance sampler (APIS) is a recent method, capable of estimating complicated multidimensional probability distributions using a population of different proposal densities. The algorithm was able to produce reasonably stable estimates for most parameters quickly. Prior to calibrating the model, we adjusted the site-specific vegetative area fractions to

reproduce the measured site level maximum of LAI. This practical approach resulted in improved ET and GPP fluxes, although we encountered difficulties in replicating these for sites with low LAI. The model parameters were optimised simultaneously for all sites without any additional site level tuning. The parameters that were most effective in the optimisation processes, were consistent for all stomatal conductance formulations.

The introduction of the $S$-function, to delay the start of the vegetation active season, has corrected the springtime increase in

GPP for conifers throughout the sites used in this study. The parameters $\theta_{tsp}$ and $\theta_{pwp}$, that set the range for the soil moisture stress function $\beta$, were both systematically lowered and optimised to nearly identical values for all stomatal conductance models. The low effective range for the $\beta$ function rendered the experimental parameter $q$ nearly ineffective in the more general optimisation. The dry period optimisation increased the effective range of the $\beta$-function and the importance of $q$, which resulted in highly nonlinear (additional) reduction in the net assimilation rate. Overall, this fact and both optimisations

indicate that boreal forest transpiration is not limited by soil moisture stress under normal conditions.

The optimisation improved the predictive skill of the model with all stomatal conductance formulations as was seen during the validation period. The Bethy, Ball-Berry and Friend and Kiang versions were the most in agreement with the observations, although the differences between these and the other formulations were small. Most of the model versions had some problems during the extended dry period and the best $b$ and $r^2$ values were achieved by the unified stomatal optimisation model. Addi-

tionally, the optimised parameter values of the USO model for the dry period were the most alike (of all stomatal conductance formulations) with those of the more general optimisation.

*Code and data availability.*   The data required to calibrate and validate the model is originally part of the FLUXNET2015 dataset that can be accessed through the FLUXNET database (doi:10.17616/R36K9X). Our modified dataset, containing the forcing data and the observations used in this article, is available through Zenodo portal (doi:10.5281/zenodo.3240954). The data depicting the simulations (parameter draws,

cost function values etc.) has been added as a supplement. The JSBACH model (branch: cosmos-landveg-tk-topmodel-peat, revision: 7384) can be obtained from the Max Planck Institute for Meteorology, where it is available for scientific community under the MPI-M Sofware License Agreement (http://www.mpimet.mpg.de/en/science/models/license/). The modifications to the model, described in this paper, have

## Appendix A: Parametric equations within JSBACH

In this appendix we present the most relevant equations that are governed by the parameters in Table 2. The appendix is divided
into sections that coincide with the parameter groups.

### A1   Photosynthesis

The Farquhar model (Farquhar et al., 1980) is based on the observation that the assimilation rate in the chloroplast is limited either by the carboxylation rate ($V_C$), induced by the Rubisco enzyme, or the light-limited assimilation rate ($J_E$). The total rate of carbon fixation is reduced by the amount of dark respiration ($R_d$), resulting in net assimilation rate ($A_n$). The experimental
scaling factor $\beta^q$ (Egea et al., 2011) is based on soil moisture stress in Eq. (2), that takes effect ($\beta < 1$) when soil moisture is significantly reduced. This scaling is used by all stomatal conductance formulations. We have also introduced here in equation form the actual reduction to photosynthesis by $\gamma$ from the delay in the start of the vegetation active season in Eq. (1).

$$A_n = \beta^q(\min(\gamma V_C, J_E) - \gamma R_d) \tag{A1}$$

Oxygenation of the Rubisco molecule reduces the carboxylation rate, which is given as:

$$V_C = V_{C,max}\frac{C_i - \Gamma_\star}{C_i + K_C(1 + O_i/K_O)} \tag{A2}$$

Here $C_i$ and $O_i$ are the leaf internal $CO_2$ and $O_2$ concentrations, $\Gamma_\star$ is the photorespiratory $CO_2$ compensation point, $K_C$ and $K_O$ are Michaelis-Menten constants parameterising the dependence on $CO_2$ and $O_2$ concentrations. Furthermore, leaf internal $CO_2$ concentration depends on the external (ambient) concentration $C_a$ (in the Baseline and Bethy formulations and unstressed conditions) by:

$$C_i = f_{C3}C_a \tag{A3}$$

Likewise, the light-limited assimilation rate can be expressed as a function on electron transport rate ($J$), which is a function of radiation intensity ($I$) in the photosynthetically active band, the maximum electron transport rate ($J_{max}$) and the quantum efficiency for photon capture ($\alpha$):

$$J_E = J(I)\frac{C_i - \Gamma_\star}{4(C_i + 2\Gamma_\star)}, \qquad J(I) = J_{max}\frac{\alpha I}{\sqrt{J_{max}^2 + \alpha^2 I^2}} \tag{A4}$$

### A2   Soil water

In JSBACH the soil water budget is based on several reservoirs (e.g. skin, soil, bare soil, rain intercepted by canopy etc.) and the different formulations are plentiful. We present here only the most crucial of these. Changes in volumetric soil water ($\theta_s$, not

to be confused with relative soil water content $\theta = \frac{\theta_s}{\theta_{fc}}$) due to rainfall ($R$), evapotranspiration ($ET$), snow melt ($M$), surface runoff ($R_s$) and drainage ($D$) are calculated with a geographically varying maximum field capacity ($\theta_{fc}$) and soil water density ($\rho_w$).

$$\rho_w \frac{\partial \theta_s}{\partial t} = (1 - p_{int})R + ET + M - R_s - D \tag{A5}$$

5      The interception parameter ($p_{int}$) also affects the amount of water intercepted by vegetation and bare soil which further affects evaporation and transpiration. The skin reservoir is limited by $w_{skin}$ and excess water is transferred to soil water. Likewise when the soil water content ($\theta$) is greater than parameter $\theta_{dr}$, the excess water is rapidly drained (in addition to the limited drainage below this threshold), where $d$, $d_{min}$ and $d_{max}$ are constant parameters:

$$D = d_{min}\theta + (d_{max} - d_{min})\left(\frac{\theta - \theta_{dr}}{1 - \theta_{dr}}\right)^d, \qquad \theta \geq \theta_{dr} \tag{A6}$$

10      Evaporation from wet surfaces ($E_{ws}$) depends on air density ($\rho$), specific humidity ($q_a$), saturation specific humidity ($q_s$) at surface temperature ($T_s$) and pressure ($p_s$) and aerodynamic resistance ($R_a$). The aerodynamic resistance depends on heat transfer coefficient ($C_h$) and horizontal velocity ($v_h$).

$$E_{ws} = \rho \frac{q_a - q_s(T_s, p_s)}{R_a}, \qquad R_a = C_h |v_h|^{-1} \tag{A7}$$

     Transpiration from vegetation ($T_v$) is likewise formulated but additionally depends on the stomatal resistance of the canopy 15 ($R_c$), which is an inverse of the stomatal conductance and as such, depends on which conductance model is used.

$$T_v = \rho \frac{q_a - q_s(T_s, p_s)}{R_a + R_c} \tag{A8}$$

     Evaporation from dry bare soil ($E_s$) also has an added dependence on surface relative humidity ($h_s$) calculated from soil dryness:

$$E_s = \rho \frac{q_a - h_s q_s(T_s, p_s)}{R_a}, \qquad h_s = \max\left[\theta_{hum}(1 - \cos(\pi\theta)), \min\left(1, \frac{q_a}{q_s(T_s, p_s)}\right)\right] \tag{A9}$$

20 The total evapotranspiration is a weighted average of $E_{ws}$, $T_v$ and $E_s$, where the weights are based on fill levels of reservoirs and the vegetative fraction of the grid cell.

## A3    Logistic Growth Phenology (LoGro-P) model

The parameters from the LoGro-P are mainly used to determine the spring and autumn events for JSBACH. To determine the date of the spring event we first introduce a few additional variables, namely the heatsum $S_T(d)$, the number of chill days $C(d)$ 25 and the critical heatsum $S_{crit}(d)$. $T(d)$ denotes the mean temperature at day $d$.

$$S_T(d) = \sum_{d'=d_0}^{d} \max(T(d') - T_{alt}, 0) \tag{A10}$$

Heatsum $S_T(d)$ cumulates the amount of "heat" above the parameter $T_{alt}$ after the previous growing season. The actual starting date $d_0$ of the summation need not be known since it is enough to start the summation "reasonably late" after the last growth season.

$$C(d) = \sum_{d'=d_a}^{d} H\left(T_{alt} - T(d)\right) \tag{A11}$$

The number of chill days is calculated as the number of days when the mean temperature is below $T_{alt}$. Here $H()$ denotes the Heaviside step function and the summation starts at the day $(d_a)$ of the last autumn event.

$$S_{crit}(d) = S_{min} + S_{range}e^{-C(d)/C_{decay}} \tag{A12}$$

The critical heatsum $(S_{crit})$ decreases as the number of chill days $C(d)$ increases, with an exponential memory loss parameter $C_{decay}$. The spring event happens when:

$$S_T(d) \geq S_{crit}(d) \tag{A13}$$

The autumn event requires the definition of one more variable, the (pseudo) soil temperature $(T_s(t))$, which at time $t$ is calculated as an average air temperature $(T)$ with an exponential memory loss $(T_{ps})$. The autumn event occurs when $T_s$ falls below a certain threshold. In the equation $N$ is the normalization constant and $\tau$ is the length of a time step.

$$T_s(t) = \frac{1}{N} \sum_{n=-\infty}^{t} T(n)e^{-(t-n)\frac{\tau}{T_{ps}}} \tag{A14}$$

## Appendix B: Stomatal conductance formulations

In this appendix we present the stomatal conductance model formulations used in this study. In the original JSBACH formulation, the Baseline model (Knorr, 1997), the photosynthetic rate is resolved in two steps. First the stomatal conductance under conditions with no water stress is assumed to be controlled by photosynthetic activity (Schulze et al., 1994). Here the leaf internal $CO_2$ concentration is assumed to be a constant fraction $(C_{i,pot} = f_{C3}C_a)$ of ambient $CO_2$ concentration $(C_a)$. This allows for an explicit resolution of the photosynthesis (Knorr, 1997). Then the impact of soil water availability is accounted for by a soil moisture-dependent multiplier $(\beta)$ that is identical for each canopy layer (Knorr, 1997).

$$g_{s,pot} = \frac{1.6A_{n,pot}}{C_a - C_{i,pot}} \quad \Rightarrow \quad g_s = \beta g_{s,pot} \tag{B1}$$

After accounting for soil water stress, the net assimilation rate $(A_n)$ and intercellular $CO_2$ concentration are $(C_i)$ are recalculated using $g_s$, and integrated over the leaf area index to produce canopy level estimates.

In the Bethy approach (Knorr, 2000), the unstressed canopy conductance $(G_{c,pot})$ is calculated similarly to the Baseline model, but potentially further limited by the water supply function of the maximum transpiration rate $(T_{supply} = \beta T_{max})$. $T_{max}$ is a fixed and predefined upper limit for transpiration as in Knauer et al. (2015).

$$G_c = \begin{cases} G_{c,pot}\frac{T_{supply}}{T_{pot}}, & T_{pot} \geq T_{supply} \geq 0 \\ G_{c,pot}, & T_{pot} < T_{supply} \end{cases} \quad , \quad T_{pot} = \rho\frac{q_s - q_a}{1/G_a + 1/G_{c,pot}} \tag{B2}$$

The potential (unstressed) transpiration rate ($T_{pot}$) is a function of air density ($\rho$), saturation specific humidity ($q_s$) at given temperature and pressure, specific humidity ($q_a$), aerodynamic conductance ($G_a$) and unstressed canopy conductance ($G_{c,pot}$). After this scaling, the net assimilation rate and intercellular $CO_2$ concentration are recalculated as in the Baseline model.

The Ball-Berry variants relate the stomatal conductance ($g_s$) to empirically fitted parameters $g_0$ (mol m$^{-2}$s$^{-1}$) and $g_1$
(unitless, except for $g_1^{USO}$ which has units of $\sqrt{kPa}$) that respectively represent the residual stomatal conductance and the slope of the function. The stomatal conductance is a function of the net assimilation rate ($A_n$), the water stress factor ($\beta$) and the atmospheric $CO_2$ concentration ($C_a$). The original Ball-Berry formulation (Ball et al., 1987) also depends on relative humidity at leaf surface ($h_s$). In the Leuning model (Leuning, 1995), the $CO_2$ concentration is reduced by the $CO_2$ compensation point ($\Gamma$) as well as scaled by the vapour pressure deficit ($D_s$) and a constant ($D_0$) depicting the stomatal sensitivity to changes in $D_s$. The Friend and Kiang model (Friend and Kiang, 2005) adds an exponential dependency on the difference of specific ($q_a$) and saturation specific humidity ($q_{sat}$) with empirically fitted constants $a = 2.8$ and $b = 80$. The unified stomatal optimisation model (Medlyn et al., 2011) also adds a dependency to the vapour pressure deficit ($D_s$).

$$g_s^{BB} = g_0^{BB} + g_1^{BB} \beta \frac{A_n h_s}{C_a} \qquad\qquad g_s^{Leu} = g_0^{Leu} + g_1^{Leu} \beta \frac{A_n}{(C_a - \Gamma)(1 + D_s/D_0)} \tag{B3}$$

$$g_s^{F\&K} = g_0^{F\&K} + g_1^{F\&K} \beta \frac{A_n a^{-d(q_{sat}-q_a)}}{C_a} \qquad\qquad g_s^{USO} = g_0^{USO} + 1.6 \left(1 + \frac{g_1^{USO}\beta}{\sqrt{D_s}}\right) \frac{A_n}{C_a}$$

*Author contributions.* The experiments were planned by J. Mäkelä, T. Aalto, T. Markkanen and T. Thum. J. Mäkelä ran the simulations and prepared the manuscript with contributions from co-authors. J. Knauer originally implemented the Ball-Berry type stomatal conductance formulations into JSBACH under S. Zaehles supervision. J. Susiluoto maintained the framework for testing the algorithm. M. Aurela, A. Black, M. Heimann, A. Lohila, I. Mammarella, H. Margolis and H. Kobayashi provided the site level observations required in this study. T. Aalto, T. Markkanen and T. Viskari extensively commented and revised the document.

*Competing interests.* The authors declare that they have no conflicts of interest.

*Acknowledgements.* This work has been supported by Jenny and Antti Wihuri Foundation, the NordForsk Nordic Centre of Excellence under Grant no. 57001 (eSTICC) and the Academy of Finland under Grant no. 295874 (OPTICA), as well as Academy of Finland Centre of Excellence under Grant no. 307331 and ICOS-Finland (project No. 281255) and EU-Life+ project MONIMET (LIFE12 ENV/FI000409). This work used eddy covariance data acquired and shared by the FLUXNET community, including these networks: AmeriFlux, AfriFlux, AsiaFlux, CarboAfrica, CarboEuropeIP, CarboItaly, CarboMont, ChinaFlux, Fluxnet-Canada, GreenGrass, ICOS, KoFlux, LBA, NECC, OzFlux-TERN, TCOS-Siberia, and USCCC. The FLUXNET eddy covariance data processing and harmonization was carried out by the ICOS Ecosystem Thematic Center, AmeriFlux Management Project and Fluxdata project of FLUXNET, with the support of CDIAC, and the OzFlux, ChinaFlux and AsiaFlux offices.

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

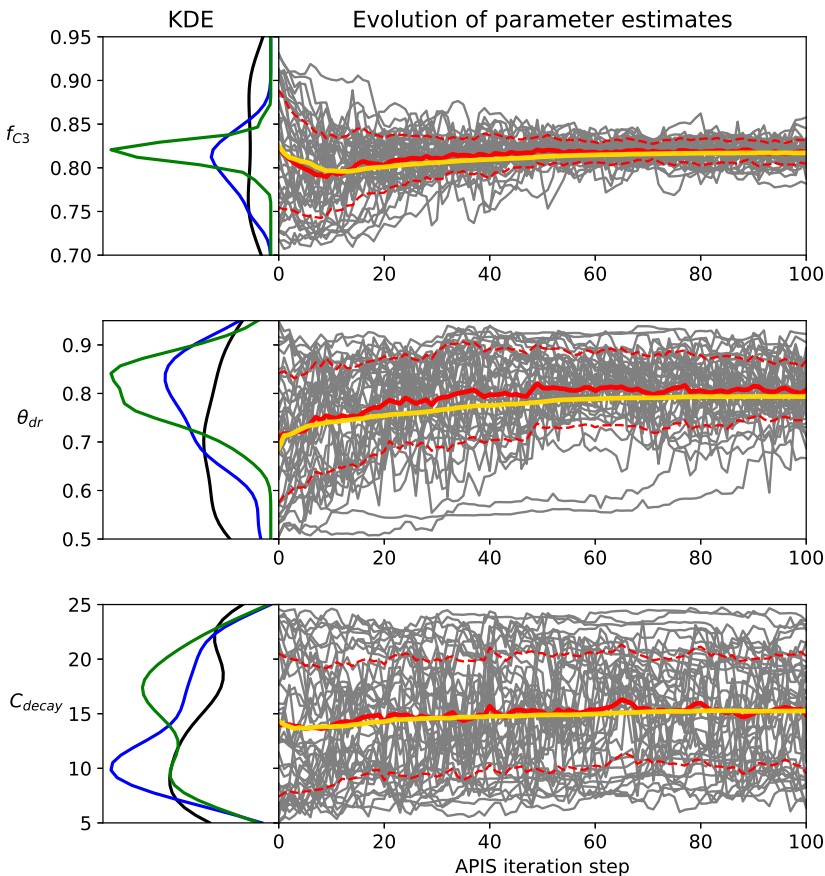

**Figure 1.** Examples of the evolution of the APIS algorithm from the Bethy calibration. The left panel is the kernel density estimate of the location parameters at the start of the process (black), after 20 iterations (blue) and after 100 iterations (green). The right panel shows the location parameters (gray), their mean (red) and one standard deviation (dashed) as well as the global estimate (yellow, calculated with the deterministic mixture approach) of the parameter expected value.

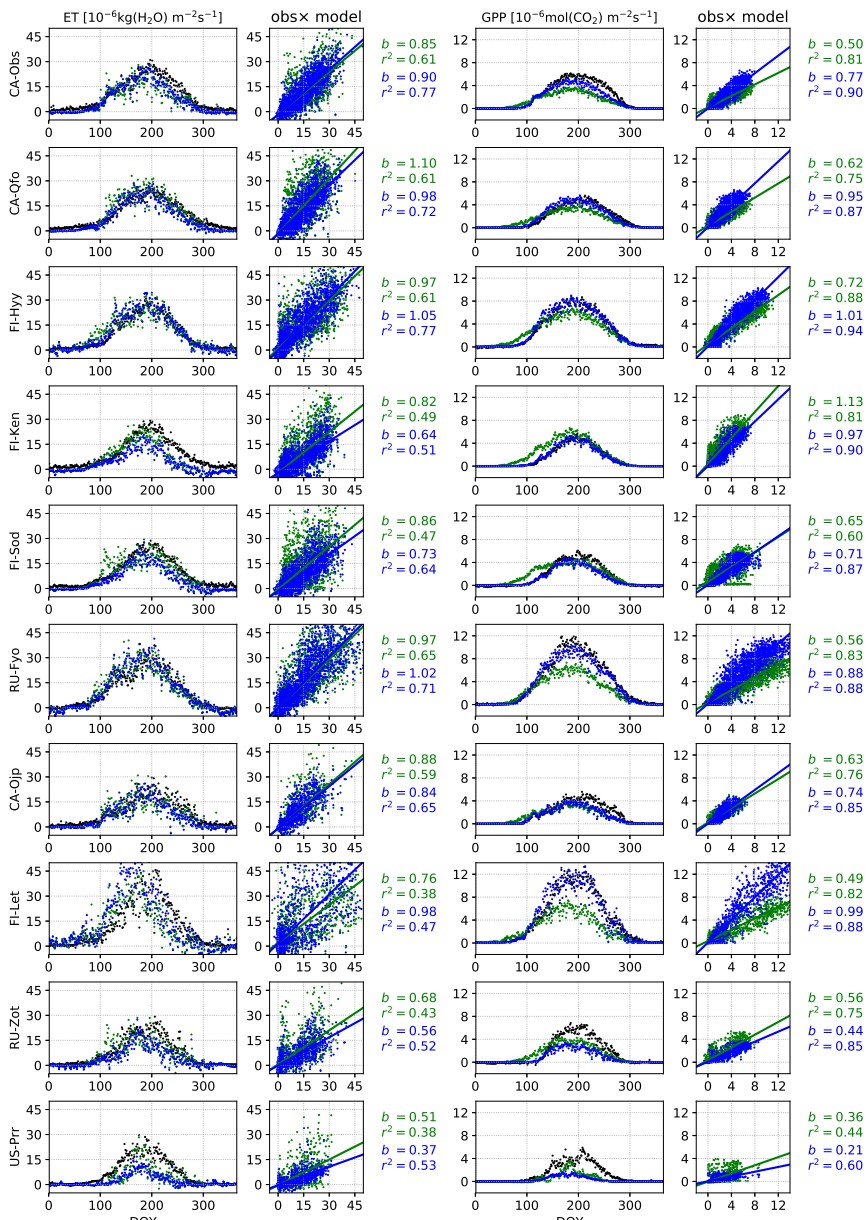

**Figure 2.** Validation period average annual cycles of evapotranspiration and gross primary production; observations (black) and the model using the Bethy stomatal conductance formulation with default (green) and optimised (blue) parameterisation. Also presented are daily model values cross plotted against observations with corresponding slope of the regression line ($b$) and the coefficient of determination ($r^2$).

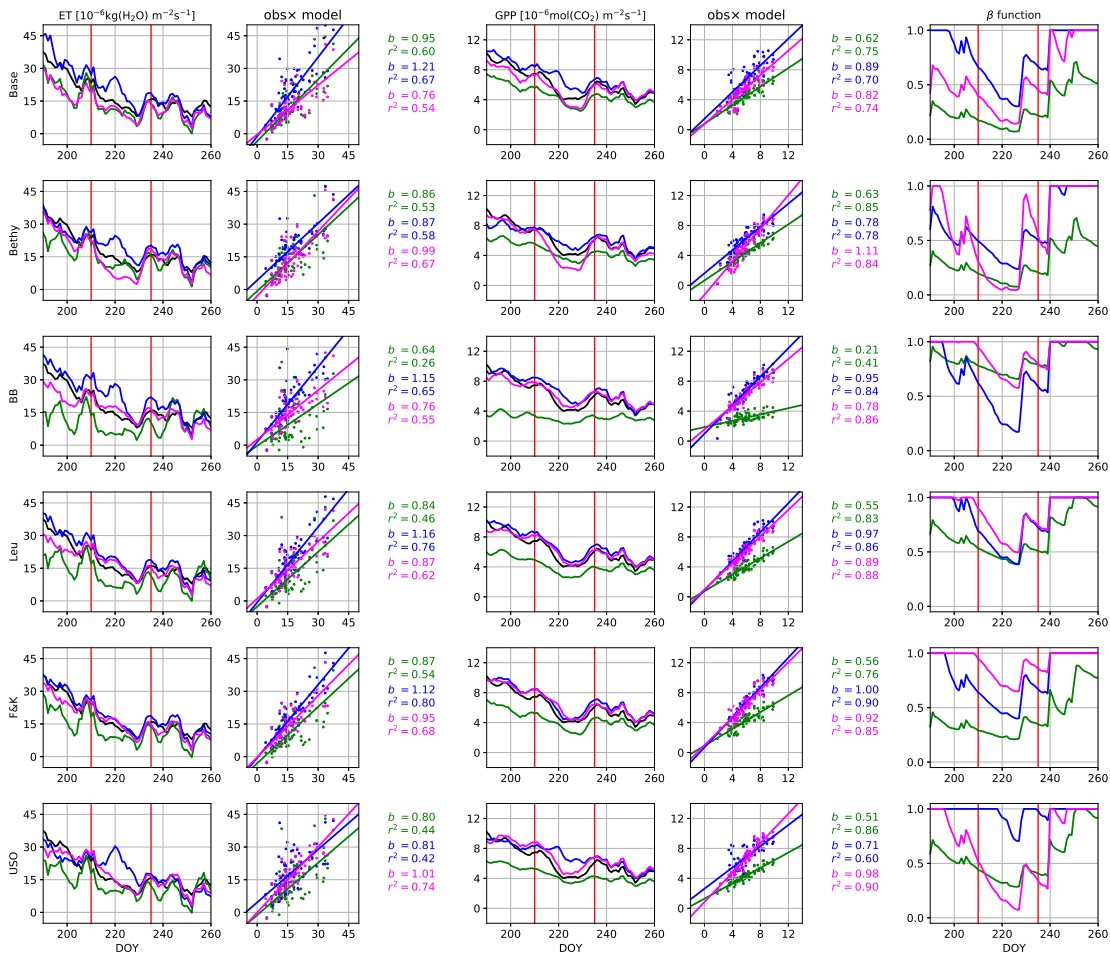

**Figure 3.** Hyytiälä site drought in summer 2006. The time series for evapotranspiration and gross primary production are 5-day running averages and for $\beta$-function daily values. The observations are plotted in black and the model with default parameterisation in green, calibration period optimisation in blue and the dry year optimisation in magenta. The red vertical lines indicate the start and end of the actual drought.

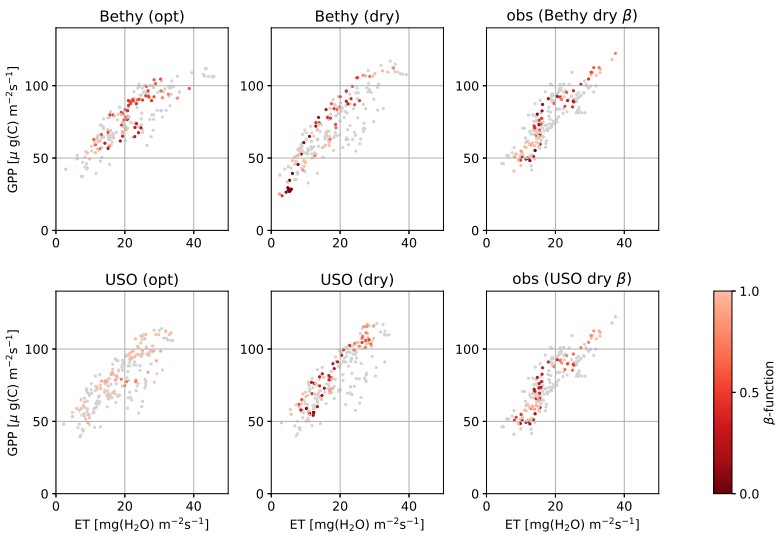

**Figure 4.** Hyytiälä site water use efficiency for the Bethy and USO formulations. Scatter plotted are the dry period 5-day running averages of ET and GPP, coloured by the intensity of the drought ($\beta$-function). The left column depicts the model with the more generally optimised parameter values, the middle column with the drought optimisation and the right column presents the corresponding observations, coloured by the same intensity as in the middle column. The grey points are from the corresponding time during the two previous years.