# Peer review of "Parameter calibration and stomatal conductance formulation comparison for boreal forests with adaptive population importance sampler in the land surface model JSBACH"

_Geoscientific Model Development, 2018_

## Short Comment (SC1) · 13 Mar 2019

Dear authors,

In my role as Executive editor of GMD, I would like to bring to your attention our Editorial version 1.1:

http://www.geosci-model-dev.net/8/3487/2015/gmd-8-3487-2015.html

This highlights some requirements of papers published in GMD, which is also available on the GMD website in the 'Manuscript Types' section:

http://www.geoscientific-model-development.net/submission/manuscript_types.html

In particular, please note that for your paper, the following requirements have not been met in the Discussions paper:

- "The main paper must give the model name and version number (or other unique identifier) in the title."

- "If the model development relates to a single model then the model name and the version number must be included in the title of the paper. If the main intention of an article is to make a general (i.e. model independent) statement about the usefulness of a new development, but the usefulness is shown with the help of one specific model, the model name and version number must be stated in the title. The title could have a form such as, "Title outlining amazing generic advance: a case study with Model XXX (version Y)"."

- "All papers must include a section, at the end of the paper, entitled 'Code availability'. Here, either instructions for obtaining the code, or the reasons why the code is not available should be clearly stated. It is preferred for the code to be uploaded as a supplement or to be made available at a data repository with an associated DOI (digital object identifier) for the exact model version described in the paper. Alternatively, for established models, there may be an existing means of accessing the code through a particular system. In this case, there must exist a means of permanently accessing the precise model version described in the paper. In some cases, authors may prefer to put models on their own website, or to act as a point of contact for obtaining the code. Given the impermanence of websites and email addresses, this is not encouraged, and authors should consider improving the availability with a more permanent arrangement. After the paper is accepted the model archive should be updated to include a link to the GMD paper."

So in your case, JSBACH must be included into the title of the manuscript. Additionally an identifier / version number indicating the exact version of the code must be

added. Note, that the code modifications you are discussing in your manuscript need to be made available. Especially note, that the exact version of JSBACH used in your manuscript needs to be permanently archived. The information how this is achieved need to be added to the code availability section.

Yours,

Astrid Kerkweg

---

## Referee Comment (RC1) · Anonymous Referee #1 · 2 Apr 2019

Overview:

1. The authors apply adaptive population importance sampler and a simple stochastic optimization algorithm to optimize the parameter sets of six different stomatal conductance models using measurements from 10 FLUXNET sites. For the validation, the experiment period is split into the optimization period and validation period at the six study sites. The remaining four study sites were used only for the validation. The reproducibility of GPP and ET was investigated with the optimized parameters. For the drought event at one flux site, the effectiveness of additional parameter optimization for water use efficiency was also investigated.

2. The results indicate that the optimization scheme presented in this paper successfully improve the estimation of GPP and ET, even for the drought event. The model was also modified to use a delayed effect of temperature for photosynthesis activity.

General comments:

Parameter optimization is essential for the model development. The methods proposed in this paper successfully optimize the parameter sets which control carbon flux or water flux. The procedure of this paper seems generally adequate, and I think the paper is relevant for GMD. However, the manuscript is needed to be improved from the two aspects:

1. The readers of this paper may not understand and reproduce the experiment because some procedures in the paper are not clear. In addition, the descriptions are sometimes too much redundant or too much simple, and the argument becomes unclear. The authors need to improve the manuscript carefully according to the specific comments.

2. The authors indicate that some of the settings are not appropriate for the USO model to simulate the drought event. Nevertheless, the authors concluded that the estimation with USO is one of the "best" results. The authors should run the experiment again with the appropriate setting if they would like to use the USO result for the discussion.

Specific Comments:

(1 is related to general comment 1, and 2 is related to general comment 2)

1-1. P2 Lines 3-8: The explanation for soil drought is confused. I think it is better to explain "general" soil drought first, and then emphasis the importance of soil draught at the boreal forests. This section is important to explain why the authors chose boreal sites for the experiment. I also do not understand the sentence "reversing the development".

1-2. P2 Lines19-20: I do not understand the sentence; "However, it can be

. . .conductance formulations."

1-3. P2 Lines 25-27: "We will assess the inter-site variability ...one site." and "We will provide an assessment of ... descriptions." are about the validation. I think it is better to explain the validation process explicitly starting with e.g. "The validity of the optimized parameters is assessed ...". The explanation should include the two points: 1) At the six flux sites, the experiment period is split into the optimization period and validation period, and the reproducibility of GPP and ET with the optimized parameters were investigated. The remaining four sites were used only for validation. 2) The drought event at one flux site is also investigated with some of the optimized (fixed)model parameters and with additional parameter optimization for water use efficiency.

1-4. P2 Lines 27-28: I think the sentence about the optimization method, "We utilize the adaptive population importance sampler ... (peak of high probability)." is too much short. One paragraph may be needed to explain this part. Many optimization schemes have been used for land ecosystem models. Therefore, it is better to review some of them, and the authors should explain the difference between APIS and well-known methods (e.g. MCMC and some other optimization methods). There are also several studies which estimated model parameters using flux site measurement. Therefore, advantages of this study also needed to be explained comparing to those previous studies.

1-5. P3 Lines 2-5: I do not understand the procedure clearly. - What is the time resolution of the model? - What is the difference between "half-hourly" and "daily" time series? How these different time series are used in the experiments?

1-6. P3 Lines 7-10: I think the explanation is confused. - Please explain the experimental setting more intelligible and clearly (refer to 1-3 of my comments), and the detailed explanations written in these sentences should be added. - "(with measurements separated into successive time period)": How many years are used for optimization and validation respectively?

1-7. P5 Table2: Definition and range of g1: "Table3" instead of "-" may be better to understand. I could not find the explanation about initial distributions for these parameters. How do the initial ranges in Table 2 relate to the black lines in Fig. 1?

1-8. P6 Lines 7-8: "However, coniferous evergreen trees do not ... following spring." I do not understand the connection before and after this sentence.

1-9. P7 (2.4 Parameter estimation) – P11 (2.9 Cost functions): I do not understand the procedure for the parameter optimization clearly. I think it is better to add the section for "overview of the experimental settings" between section 2.3 and section 2.4. Then, Section 2.7 may better to be included in this overview section. The procedures for initial parameter settings, spin-up, parameter optimization, and validation should be easily understood. A process diagram may help the readers to understand.

Especially, I do not understand the relations between the "parameter estimation" by adaptive population importance sampler (section 2.5) and "parameter optimization" by the simple custom stochastic optimizer (section 2.6). - In my understanding, the APIS is used to estimate model parameters roughly. Then using the estimated distributions by APIS as the initial state, parameter optimization is done. Is that correct? The overview section should include more detailed explanation for this point. Introduction (P1 Lines 2-3) also needed to be corrected so that the readers can understand the procedure. - How many years observations are used for these two different optimization methods? Are the observations for the two optimization methods same? - I do not understand how to merge the optimizes parameters as shown in Table4, because in my understanding, the parameters are optimized separately at each study site.

It is also necessary to clarify the role of "2.8 Simulation analysis" and "2.9 Cost function" at the overview section. In my understanding, the first half of 2.8 indicate the parameter sensitivity, and the latter half of 2.8 indicates the validation of the result using the observation. Cost function (2.9) is used both for parameter optimization (APIS and a simple stochastic optimizer) and for validation of the results (e.g. Table 4, Table
6). These descriptions also should be included in the overview.

1-10. P8 Line 23: What is "M=40 proposals"?

1-11. P8 Line 30 – P9 Line 3: I do not understand the procedure (see, 1-9 of my comment).

1-12. P9 Lines 30-33, P10 Line2: I do not understand these sentences (see, 1-9 of my comment).

1-13. P11 Line 29: What is "sampling limits set"?

1-14. P12 Lines 1-18: This may be better explained in introduction (see my comment 1-4). The parallel mode is not used in this study, therefore this advantage (parallel simulation) is not suitable for this study setting.

1-15. P12 Lines 29-31: "The actual soil moisture ... unreliable and even unrealistic.": Then, what is the recommended setting for the future study? Is it OK for this experiment?

1-16. P14 Line 25 – P15 Line 2: Some descriptions are redundant. Improvement of description is needed so that the readers understand the Fig. 2 clearly.

1-17. P16 Line 9: "We optimized the model for individual (calibration) sites as well.": I do not understand "as well". I thought the model was optimized at the individual sites.

1-18. P16-P19: I do not understand the arguing point in this section (3.3). I think it is better to explain Table 6, Fig.3 and Fig. 4 first, and then more detailed discussion should be done. - P16 Lines 26-32: Too much detailed and complicated. First, the categorization of the optimized (fixed) parameters and the parameters for further optimization (for WUR) should be explained using Table 4 and Table6. What is the most important different between these parameter groups? The detailed settings for the fixed parameter may better be explained in Appendix. - P16 Line 33 - P17 Line 4: This paragraph is important to describe the parameter optimization for the drought event.

How many years WUR optimization was done? Is the optimization procedure different only for cost function calculation? Are the observations for year 2006 repeatedly used?

1-19. P17 Line 16-18: The parameters are just optimized in this experimental setting, and the "true value" is not known. Therefore, I think "optimal value" should not be used here. Authors can just say that "the optimized parameter set for WUE greatly improved the simulation results (Fig. 3)".

1-20. P17 Line 14 – P 18 Line 18: The detailed explanations for each parameter are too much complicated and I do no understand. The paragraph of P18 Line 19-26 should be placed before these paragraphs. Then, the relationship between the results in Fig. 3 and the estimated parameters should be discussed as below: - Which parameters are the important to control WUE in this experiment? - How do these parameters affect WUE? - Are the estimated parameters reasonable compared to the previous studies? If not, why?

1-21. P18 Lines 22-23: I do not distinguish "the actual drought" in Fig. 3. I think it is better to add the period of the drought in the Fig. 3.

1-22. ïijř18 Lines 27-33: I do not understand this paragraph because I do not understand Fig. 4. Does the lower panel show USO results? What is "Medlyn"? I also do not understand how is the $\beta$-function for the observation calculated.

1-23. P18 Lines 34- P19 Line 5: The authors did not show the experimental setting and result explicitly, therefore I do not understand the purpose of this experiment. What is the difference between this experiment and the parameter optimization in section 3.3?

1-24. P19 Lines 17-20: I do not understand these sentences.

2-1. P17 Lines 7-13: The authors explain that the setting for USO is not appropriate. Then the results should not be used for further discussion after this paragraph. If authors would like to use the result, they should perform the experiment again with the appropriate settings.

[Figure]

Technical corrections:

1. P1 L11: "correctly time and replicate" -> "correctly reproduce"

2. P2 Line 27: Abbreviation "APIS" should be placed here (this is the first appearance).

3. P9 Line 5: "from 2006" -> "in 2006" (only one-year optimization).

4. P10 Lines 17-18: "high", "average", or "low" effectiveness value: this explanation should be the same as Table 4.

5. P11 Lines 19-20: Description of the Supplemental materials is needed at under each figure.

6. P15 Table5: some values are different form the Fig. 2 (i.e., r2 of Bethy).

7. P18 Line13: "disregardin" -> "disregarding"

8. P17 Line 14 "The most noteworthy" what? (modified word is needed)

---

## Referee Comment (RC2) · Anonymous Referee #2 · 3 Apr 2019

Dear editors, dear reviewers,

I enjoyed reading this study by Mäkelä on the calibration of a new version of the JS-BACH model. I find that topic and general approach fit well to Geoscientific Model Development, and that the paper has the potential to make an informative and useful contribution in this field.

That being said, I currently see two major problems with the study (detailed below), as well as a number of smaller issues that need to be cleared up before publication.

MAJOR / GENERAL ISSUES

1) The current abstract, and much of the method section, are concerned with the calibration of the model. At the same time, however, the authors make several modifications of the model (which are mostly described in the appendix), apparently in response to shortcomings that were identified in earlier studies, and present an additional case study (summer droughts in Hyytiälä) to demonstrate the improved properties of the model. As a reader, one gets the feeling that at least two studies were combined in one: i) a study about the calibration of JSBACH, with side notes about the effectiveness of the APIS algorithm ii) a study about model improvements. The case study on Hyytiälä seems to me, with due respect, a bit of a Finnish obsession – many models have problems with properly reproducing flux characteristics of Hyytiälä, I guess due the somewhat unusual soil / climate combination of this site, and although it's nice that the improved model fares better than the previous version, I'm not sure if there is a scientific reason for giving so much space to the performance of this site for a general vegetation model. To address this entire point, I would urge the authors to consider my comments, and think about whether this paper could / should be restructured, possibly by giving more space to model improvements in intro / methods.

2) The calibration procedure has several severe technical shortcomings that should be resolved. The most important is the formulation of the likelihood, which, at the moment, is not a real likelihood, but just an arbitrary cost function. You could also keep it like that, but then you shouldn't call the result posterior (as it is not based on a reasonable estimate of p(D|M)). Moreover, you should provide convergence diagnostics. If you want to make strong statements about the quality of APIS, I would recommend a benchmark against a suitable algorithm for you problem (I make a recommendation later). Moreover, I did not understand why an optimization is necessary on top of the posterior estimation. The MAP could also be estimated from the posterior sample (unless high precision about the exact location of the MAP is required)

3) Again, a bit of a broad comment, but I found that many of the conclusions are only weakly supported by the results, and also in the results and discussions, there are of

points of interpretation that seem only weakly linked to the results. Could you please make sure, throughout the manuscript, that the discussion concentrates on tangible, numerical results, and that it is clear to the reader on what results you base your interpretation (e.g. by appropriate references to tables, figures, SI)

DETAILED COMMENTS

Title: says nothing about model improvements

1.3 Name algorithm

1.7 this sounds very vague – how was performance compared, and why do you say on the one hand that there was no clear best model, and then that some models were better.

1.8 why would the improvement in the Finnish site be important?

1.10 This seems a completely unconnected question that is suddenly introduced here at the end of the abstract. See main comment 1

2.12 logical gap here – not clear why the problems named before call for species / zone specific parameterizations.

2.19 Why can this be hypothesized? To me, the only logical reason is that all models have (different) model errors, which is thus always compensated (differently) by other parts of the model. Is that your logic? If so, please make this explicit

2.25 It does not become clear why it would be necessary to study inter-site variability or the specific drought even for the questions you have posed before

2.29 This seems to contradict the abstract, where you state you use an optimizer for the optimum. But you could of course have estimated the mode via a suitable density estimator, or just take the parameter value with highest posterior within the sampled points.

[Figure]

2.30 I'm not sure why you provide this information at this point – an overview about the methods would have been more logical

3.3 by aggregates you mean means?

3.8 The temporal split is of course a less independent validation than a new site, but OK, why not . . .

4.3 What do you mean by "observational metereological dataset" – where the weather stations on the flux sites? If not, how far away, and is that a problem?

4.5 I'm not really sure why you would want to consider a feedback in this context, i.e. if you have climate measurements on site. Probably relates to previous question

4.8 Why do you need two citations for that fact that you don't work on a grid, but plot-based, i.e. for what are those references cited?

4.15 "fractional structure"? I think the first part of the sentence is just clutter, just say: In JSBACH, the land surface is divided into grid cells, and the grid cells are divided into tiles . . .

4.17 site-level

4.21 Although this seems logical on the first glance, it's not always clear if the "right" LAI setting for a model is the measured LAI, because models often assumes homogenous leaf distributions, but real leaf distribution is inhomogeneous, a lower-than-measured LAI will sometimes produce more appropriate photosynthesis values (cf Medlyn, Belinda E. "Physiological basis of the light use efficiency model." Tree physiology 18.3 (1998): 167-176.). It depends on the model structure. I wonder if this would better be calibrated as well, or at least I'd like to hear your comments about the assumptions about leaf distribution in JSBACH and if setting observed LAI is clearly appropriate.

5.2 You give no reasons, but I assume the modifications were done to facilitate the calibration?

5.7 Why give numbers for the groups and not a name?

5.5. The sentence is unintelligible. Moreover, the explanation of how parameter ranges (i.e. priors – why don't you call them priors) are derived is not sufficient. Provide a clear rationale for prior elicitation.

6.1 "The" lacking. In general, you are very economic regarding the use of articles.

6.3 heatsum sum

6.6 ready

7.11 You didn't define Chi, but I assume this is your prior space? Also, there is no need that this space is a subset of R^n (you can have discrete parameters)

7.11 Likewise, observations don't have to be continuous, thus not element R

6.15 What do you mean by directly assessed?

eq 3: the sense of the three different formulations of the right side of the formula evades me. The middle one is Bayes formula, the other two seem nonsense. If you want to define p(x|y) as l(x|y), why not define this directly. Moreover, usual notation for Likelihood is curly capital L. Same for g(x) – why first introduce the prior as p(x) and then rename it to g(x)? I also see no need for Z – if we keep on writing p(y), eq. 4 is much easier to understand

7.25 It is a VERY unorthodox notation to define pi(x) as the posterior pi(x) is often used for the prior, to distinguish it from p(x|D). I found this highly confusing

eq. 4 This seems to me a crazy reformulation of the formula, as it is so much harder to see why this holds as if you would just write the standard p(D) = int p(D |x) p(x) dx, which shows that if you marginalize the posterior over the space X, you are left with p(D).

8.30 This entire procedure remained nebulous to me. First of all, if you favor the proposal mean, you should correct this in the acceptance probabilities, right?. How was this done? Secondly, when I understand correctly, you use the same spin-up (from the mean) for all parameters? I don't see how this can be justified, and how this could be corrected. What does "slightly scale" mean, do you increase or decrease weights?

9.9 If I understand correctly, you developed a new optimizer here? Why not use a well-known, tested optimization algorithm? In general, what you do here looks like a pretty standard gradient descent method. I would suggest to re-run this with an established optimizer (apart from the fact that I don't understand why you need an optimizer)

9.20 I have many doubts whether this algorithm makes sense / performs better than alternatives, and would recommend to test optima against a reliable algorithm (DEoptim package in R is very reliable for complicated target in my experience), but OK, it's probably not the main point about this paper. I just don't understand why you wouldn't fall back on standard solutions wherever possible.

10.4 The spin-up procedure seems to favor the proposal mean and could thus distort the posterior. Please discuss

10.9 I'm not sure if I understand correctly – you are applying a KDE on the sampled posterior, and then create samples from that for the posterior predictive distribution? Why would you do that? Would that (potentially) distort the posterior? Please discuss and if you do what I think you do, prove that this does not distort the posterior.

10.13 What's the sense of this effectiveness? It seems this is something like sensitivity, but this could be calculated directly from the difference prior – posterior. Moreover, when I understand correctly, this is a kind of conditional calculation, where you keep the other parameters at the optimum? Makes kind of sense, but is also loosing info about the parameter correlations in the posterior, so in theory, a parameter could be very "effective", despite being globally poorly constrained (due to a trade-off with another parameter). Please discuss.

10.25 Based on your exposition, you should define a likelihood, and not a cost function. This word has no meaning in a Bayesian context.

10.25 If this is a likelihood, correct interpretation would be that likelihood is normal

10.26 a) what do you mean by "successfully done" – that it went through the review? please give a reason for this b) so, the cost function is NOT the MSE

10.29 Not clear to me what you mean by "covariance vector", and "combining model and observation error". You don't know the model error as such, but of course, as in a linear regression, you can fit the sd of the normal distribution to the effective spread around the predicted value, which is the standard approach.

eq 9 – OK, if you want to define this as your likelihood, then you should simply state the correct assumptions. What you assume here is that the relative error is normally distributed, with a standard deviation EQUAL the mean, and additionally you divide the likelihood by the number of data points (dividing by N_ET), which makes no sense if you truly want to create a likelihood. Why does this not make sense?

a) you don't know a priori how your residual scatter around the model predictions. The scale of the normal distribution (essentially the denominator in the likelihood) affects the shape of the posterior, i.e. makes it more or less wide. As you can't know the correct scale, you have to fit a parameter here

b) also, it doesn't make sense to have residual go to zero for small observations. A sensible expression for the scale of the normal would be to fit

log likelihood = (predicted – observed / (a0 + a1 * predicted))^2

where parameters a0, a1 have to be optimized.

c) there is no good statistical reason to divide by N_ET, i.e. by using a mean squared error as the likelihood. Essentially, by doing this, you scale the likelihood to have the evidence of one data point, making the posterior much wider than it would naturally be

[Figure]

In general, it seems to me that the likelihood you use here creates a posterior that is far wider than any sensible statistical assumptions would allow.

11.8 I found the structure with results and discussion together not very helpful. It seems to me that this is adding to the fragmentation of this paper, which seems to address several questions (model improvement, calibration, drought case study) at the same time. A discussion which summarizes the results and puts them in a common perspective would have seemed preferable to me

11.13ff it seems you suggest in this paragraph that identifiability equals or is related to convergence, and it's not clear to me why (in general, these are two different issues). Moreover, I can see no visual difference in convergence speed between the three examples. If you claim the first one converges faster, please back this up by numeric estimates of convergence, e.g. Gelman-Rubin.

11.13 Moreover, you should provide convergence diagnostics / proof of convergence (typically Gelman-Rubin) for all your results! Not having checked convergence is not acceptable.

11.16 It's not the algorithm that is unable to constrain the parameter, it's the likelihood. Also, you seem to suggest that this is a problem, but that's perfectly normal for a Bayesian analysis.

11.20 What does reasonably stable mean? See comment about convergence diagnostics above

11.24 Again, not really clear to me why you do the optimization in the first place, instead of estimating the MAP from the posterior sample

11.25 Near or at. Why is near the limits a problem? If you have flat priors, you state that all these values are equally likely, so near limit is no problem. I suspect though that you have MAPs at the limit, posterior medians are of course never at the limit.

2.8 You can parallelize the chains in DREAM, which means that, assuming you run 3

MCMCs as usual, you can use at least 9 cores. I'm not sure how many cores you were using. On the plus side, I'd bet that DREAM or DEzs algorithms converge faster than APIS. I think it would be useful to benchmark against one of these algorithms. Both are implemented in the R package BayesianTools.

12.20 So, why not calibrate them right away?

12.20 / 12.26 Most of the info in these paragraphs are not results

16.22 The entire section reads like an independent case study with its own methods, results and discussion.

19.8 Comments about APIS: I could not see a serious evaluation of the convergence and quality of this algorithm in the paper. At least, you should provide convergence checks. If you want to say anything about the quality of APIS, I think you should compare to a reasonable referece. For example, DEzs or DREAMzs in the R package BayesianTools would be suitable reference algorithms that have proven to work well for these kinds of problems.

19.10 Define successful.

19.10 General comment: for any claim you make in this section about your findings, please refer to a specific result in section 3 that is the basis of your claim. Specifically, I can see no results that provide hard support for your first two claims.

19.28 Code and data availability is insufficient. Unless there is a good reason against this, please provide all code and empirical data (drivers and calibration / validation) with the paper, or in an appropriate repository. FLUXNET could be updated or changed, and in any case, it would be more convenient for the reader to have your entire data set at hand. Moreover, you should ensure computational reproducibility, but storing random seeds etc. for the algorithms. Ideally, also results, in particular MCMC chains should be saved, if space permits this.

---

## Author Comment (AC1) · 2 May 2019

**General author comments**

This document contains the executive editor and reviewer comments, and our responses to these comments, regarding the discussion of: https://www.geosci-model-dev-discuss.net/gmd-2018-313/

The editor and reviewer comments are shown here as indented text. We have added numbering to certain comments, so they may be more directly referenced, if needed. Our responses are shown after each comment as unindented text (such as this).

Based on the comments, it seems that the APIS algorithm description was not clear enough and needs both clarification and revising. We will add a comparison to a basic multichain MCMC, to highlight the differences. We will add information about the convergence of the algorithm, as discussed in more detail when answering the specific comments. We will also separate the "Results and discussion" into two different sections.

The main differences between APIS and a basic multichain MCMC sampler will be outlined in the methods section along the following lines:

- 1. In our simulations APIS is set up as 40 simultaneous, independent IS samplers that have their own prior distributions and locations. This is similar to a (simultaneous, independent) 40-chain MCMC sampler, where each chain corresponds to one IS sampler. The priors in our simulations are truncated Gaussian distributions, with initial locations randomly sampled from a uniform distribution that is defined by the ranges of each parameter. The deviations were also randomised.
- 2. At each iteration (this is called an epoch, but we have tried to avoid the use of the name), we take 50 draws with each of the 40 IS samplers (altogether 2000 draws). In basic MCMC we would take one draw for each chain (40 altogether).
- The location and shape parameters for each individual IS sampler is then updated (based on their "own" draws) – in APIS the new values are automatically accepted (this is called blind adaptation).
   MCMC chains, in contrast, would deal with acceptance probabilities etc. and accept/reject the draw accordingly (also possibly adapt the prior distribution at certain intervals).
- 4. Additionally, the APIS global estimates of the parameter expected values are updated (using deterministic mixture and all samples).

Each individual IS sampler (at 2) generates a sample of 50 draws which, reweighted according to the cost function, form an estimate of the posterior distribution. The parameter distributions, that we have presented, are formed from the location parameters of these individual IS samplers. They do not represent the \_true\_ posterior distributions, rather they are a collection of estimates of the mean. These values are expected to be around all the modes of the target. The deterministic mixture ensures the stability of the estimation of the parameter (global) expected values (4 above).

We originally finished the APIS simulations about 9 months ago. At that time we had ran the algorithm for all conductance formulations for the duration of 100 epochs. Afterwards we discovered a coding error in the Ball-Berry conductance formulations (affecting all of these models). As a result of this error, these simulations were wrong. They were, however, very indicative of the parameter distributions. After the error was corrected, we ran the APIS sampler for the BB model and verified that the resulting distributions were similar to the ones before. Because of this, we did not run the APIS for the other models, but ran the optimiser from the end state. For the sake of comparability, the optimiser was also run for Baseline and Bethy simulations.

In the APIS descriptions we followed the same notation as in the paper originally describing the algorithm. This was done, so it would be easier for readers to refer to this paper. Since this seems to be only confusing, we will revert to the more common notation (when applicable).

**Executive editor comment**

• So in your case, JSBACH must be included into the title of the manuscript. Additionally an identifier / version number indicating the exact version of the code must be added. Note, that the code modifications you are discussing in your manuscript need to be made available. Especially note, that the exact version of JSBACH used in your manuscript needs to be permanently archived. The information how this is achieved need to be added to the code availability section.

The JSBACH model is version controlled (svn) and the information about the model branch and version number will be added to the code availability section. However, we cannot grant access to the model, as it is under the Max Planck Institutes License agreement. Our modifications have been made mostly within the existing model structure, so the separation of original model and our modifications is not meaningful. We will archive our modifications e.g. in GitHub. Accessing these modules will require acceptance of the MPI-M License agreement (as the modifications are made within the model code), after which access to the modified modules can be granted on request.

**Comments by Reviewer 1**

**Overview:**

The authors apply adaptive population importance sampler and a simple stochastic optimization algorithm to optimize the parameter sets of six different stomatal conductance models using measurements from 10 FLUXNET sites. For the validation, the experiment period is split into the optimization period and validation period at the six study sites. The remaining four study sites were used only for the validation. The reproducibility of GPP and ET was investigated with the optimized parameters. For the drought event at one flux site, the effectiveness of additional parameter optimization for water use efficiency was also investigated.

The results indicate that the optimization scheme presented in this paper successfully improve the estimation of GPP and ET, even for the drought event. The model was also modified to use a delayed effect of temperature for photosynthesis activity.

**General comments:**

Parameter optimization is essential for the model development. The methods proposed in this paper successfully optimize the parameter sets which control carbon flux or water flux. The procedure of this paper seems generally adequate, and I think the paper is relevant for GMD. However, the manuscript is needed to be improved from the two aspects:

- The readers of this paper may not understand and reproduce the experiment because some procedures in the paper are not clear. n addition, the descriptions are sometimes too much redundant or too much simple, and the argument becomes unclear. The authors need to improve he manuscript carefully according to the specific comments.
- The authors indicate that some of the settings are not appropriate for the USO model to simulate the drought event. Nevertheless, the authors concluded that the estimation with USO is one of the "best" results. The authors should run the experiment again with the appropriate setting if they would like to use the USO result for the discussion.

We agree with the first general comment and will improve the manuscript. The second comment we disagree with as our message of how appropriate the situation is, may have come across too negatively. In the simulations, one of the parameter bounds for the USO model does not properly reflect the values in the literature reference (Medlyn et al. 2011). This is true for both optimisations, not only for the drought. This

parameter is not optimised to the boundary (in either optimisations). Considering also, that the "same" parameter (g1) for the other conductance formulations has been unimodal, we feel that rerunning these simulations with a smaller lower bound is not worth the effort.

**Specific Comments:**

(1 is related to general comment 1, and 2 is related to general comment 2)

1-1. P2 Lines 3-8: The explanation for soil drought is confused. I think it is better to explain "general" soil drought first, and then emphasis the importance of soil draught at the boreal forests. This section is important to explain why the authors chose boreal sites for the experiment. I also do not understand the sentence "reversing the development".

We will re-order the sentences describing the global and Boreal relevance of soil moisture drought. "Reversing the development" connects to the start of the sentence and to the "recovery of photosynthesis". We will reformulate these sentences.

1-2. P2 Lines 19-20: I do not understand the sentence; "However, it can be ... conductance formulations."

We hypothesise, that the choice of conductance model will affect the optimal values of the parameters that are not conductance model specific (so it is not enough to only optimise conductance model parameters). For example the maximum carboxylation rate (Vc,max) has different optimised values, when using the different conductance formulations. We will amend the sentence.

1-3. P2 Lines 25-27: "We will assess the inter-site variability ...one site." and "We will provide an assessment of ... descriptions." are about the validation. I think it is better to explain the validation process explicitly starting with e.g. "The validity of the optimized parameters is assessed ...". The explanation should include the two points:

1) At the six flux sites, the experiment period is split into the optimization period and validation period, and the reproducibility of GPP and ET with the optimized parameters were investigated. The remaining four sites were used only for validation.

2) The drought event at one flux site is also investigated with some of the optimized (fixed) model parameters and with additional parameter optimization for water use efficiency.

These comments are more related to the methodology of the paper, not the introduction, and we will take them into account in the proper section of the paper.

1-4. P2 Lines 27-28: I think the sentence about the optimization method, "We utilize the adaptive population importance sampler ... (peak of high probability)." is too much short. One paragraph may be needed to explain this part. Many optimization schemes have been used for land ecosystem models. Therefore, it is better to review some of them, and the authors should explain the difference between APIS and well-known methods (e.g. MCMC and some other optimization methods). There are also several studies which estimated model parameters using flux site measurement. Therefore, advantages of this study also needed to be explained comparing to those previous studies.

We have moved here the comparison to other algorithms, from the start of "Results and discussion". These paragraphs will be also modified to better suit the new context.

1-5. P3 Lines 2-5: I do not understand the procedure clearly. - What is the time resolution of the model?
- What is the difference between "half-hourly" and "daily" timeseries? How these different time series are used in the experiments?

The difference between the time series is that the other consists of measurements at every 30 minutes and the other consists of daily values. The model does not have a fixed time resolution, and it can be run with both datasets. We used the half-hourly dataset to quality check the daily values (as described in the paper). The model is run with the half-hourly dataset, but we examine the output (GPP and ET) at the daily level. We will modify the text accordingly.

1-6. P3 Lines 7-10: I think the explanation is confused. - Please explain the experimental setting more intelligible and clearly (refer to 1-3 of my comments), and the detailed explanations written in these sentences should be added. - "(with measurements separated into successive time period)": How many years are used for optimization and validation respectively?

Detailed information will be added.

1-7. P5 Table2: Definition and range of g1: "Table3" instead of "-" may be better to understand. I could not find the explanation about initial distributions for these parameters. How do the initial ranges in Table 2 relate to the black lines in Fig. 1?

Changed the "-" to "Table 3". The ranges are the absolute minimum and maximum values for the parameters in these simulations. The initial locations for each IS sampler is uniformly sampled from within this range of values. The black lines correspond to the initial distribution of the IS sampler initial locations.

1-8. P6 Lines 7-8: "However, coniferous evergreen trees do not ... following spring." I do not understand the connection before and after this sentence.

The spring event in the model determines when leaves can start to grow. Conifers do not shed all of their leaves (needles), so this "spring event" timing is not that important for them. They already possess considerable amount of leaves in spring when temperature rises and plants start to photosynthesise. In addition, the start of the photosynthetically active season for conifers in JSBACH has been observed to occur too early. So we try to amend the situation by adding the delayed effect for photosynthesis. We will amend the sentences to be clearer.

1-9. P7 (2.4 Parameter estimation) – P11 (2.9 Cost functions): I do not understand the procedure for the parameter optimization clearly. I think it is better to add the section for "overview of the experimental settings" between section 2.3 and section 2.4. Then, Section 2.7 may better to be included in this overview section. The procedures for initial parameter settings, spin-up, parameter optimization, and validation should be easily understood. A process diagram may help the readers to understand.

Especially, I do not understand the relations between the "parameter estimation" by adaptive population importance sampler (section 2.5) and "parameter optimization" by the simple custom stochastic optimizer (section 2.6). - In my understanding, the APIS is used to estimate model parameters roughly. Then using the estimated distributions by APIS as the initial state, parameter optimization is done. Is that correct? The overview section should include more detailed explanation for this point. Introduction (P1 Lines2-3) also needed to be corrected so that the readers can understand the procedure. - How many years observations are used for these two different optimization methods? Are the observations for the two optimization methods same? - I do not understand how to merge the optimized separately at each study site. It is also necessary to clarify the role of "2.8 Simulation analysis" and "2.9 Cost function" at the overview section. In my understanding, the first half of 2.8 indicate the parameter sensitivity, and the latter half of 2.8 indicates the validation of the result using the observation. Cost function (2.9) is used both for

parameter optimization (APIS and a simple stochastic optimizer) and for validation of the results (e.g. Table 4, Table 6). These descriptions also should be included in the overview.

We will add an overview section comparing APIS procedure to a general MCMC chain with a bit more in detail than the one at the start of this document. Essentially you have understood the process correctly – APIS is used to estimate the parameter posterior distribution and the stochastic optimiser starts from there. Both methods use the same data for driving and optimising the model. In the general optimization, we take the mean of the site level cost function values and use this for APIS and the optimiser (for each conductance model separately; the parameters reflecting these are reported in Table 4). The dry period optimisation uses only data from Hyytiälä 2006 and the corresponding values are presented in Table 6.

1-10. P8 Line 23: What is "M=40 proposals"?

The number of IS samplers in APIS. We will add this information to the overview.

1-11. P8 Line 30 – P9 Line 3: I do not understand the procedure (see, 1-9 of my comment).

At each iteration, we run the spin-up for the model only for the IS sampler mean (and use the same spin-up for the other 49 samples). This is done to speed up the process. This information will be added to the overview.

1-12. P9 Lines 30-33, P10 Line2: I do not understand these sentences (see, 1-9 of my comment).

This is essentially the same thing as above. The spin-up is needed to get suitable starting conditions for the model by driving some state variables (e.g. soil water content and LAI) it into a (semi) steady state. Since in APIS we are mostly interested in the means of each IS sampler, we use the same spin-up for the other samples.

1-13. P11 Line 29: What is "sampling limits set"?

The quotation should be "sampling limits set for each parameter". These are the limits (the range) given in Table 2. We have modified the sentence to better reflect this.

1-14. P12 Lines 1-18: This may be better explained in introduction (see my comment 1-4). The parallel mode is not used in this study, therefore this advantage (parallel simulation) is not suitable for this study setting.

These paragraphs have been moved to the introduction. The optimisation has been run in parallel mode and we consider this critique to be unqualified.

1-15. P12 Lines 29-31: "The actual soil moisture ... unreliable and even unrealistic.":Then, what is the recommended setting for the future study? Is it OK for this experiment?

These sentences refer to the optimisation of the parameter q in the general setting. It takes effect only with very low soil moisture values that occur rarely in the optimisation, hence the amount of data affecting the optimisation of q is very limited. Therefore any values of q should be viewed with reservations. The situation is slightly improved during the dry period optimisation, but the differences in q with different conductance formulations are quite large.

1-16. P14 Line 25 – P15 Line 2: Some descriptions are redundant. Improvement of description is needed so that the readers understand the Fig. 2 clearly.

We will separate the "Results and discussion" into two sections, which should improve the readability.

1-17. P16 Line 9: "We optimized the model for individual (calibration) sites as well.": I do not understand "as well". I thought the model was optimized at the individual sites.

The general optimisation was done for all sites simultaneously in order to preserve the generality of the model (on Boreal coniferous forests). We will try to make this explicitly clear.

1-18. P16-P19: I do not understand the arguing point in this section (3.3). I think it is better to explain Table 6, Fig.3 and Fig. 4 first, and then more detailed discussion should be done. - P16 Lines 26-32: Too much detailed and complicated. First, the categorization of the optimized (fixed) parameters and the parameters for further optimization (for WUR) should be explained using Table 4 and Table 6. What is the most important different between these parameter groups? The detailed settings for the fixed parameter may better be explained in Appendix. - P16 Line 33 - P17 Line 4: This paragraph is important to describe the parameter optimization for the drought event. How many years WUR optimization was done? Is the optimization procedure different only for cost function calculation? Are the observations for year 2006 repeatedly used?

We will take these into consideration when we rewrite the discussion. The order will be affected by the division of "Results and discussion" into two separate sections.

1-19. P17 Line 16-18: The parameters are just optimized in this experimental setting, and the "true value" is not known. Therefore, I think "optimal value" should not be used here. Authors can just say that "the optimized parameter set for WUE greatly improved the simulation results (Fig. 3)".

The "true" value is likely never known. Every optimisation experiment is situational and must be considered in the context of the optimisation. We will take the revised sentence into consideration.

1-20. P17 Line 14 – P 18 Line 18: The detailed explanations for each parameter are too much complicated and I do no understand. The paragraph of P18 Line 19-26 should be placed before these paragraphs. Then, the relationship between the results in Fig.3 and the estimated parameters should be discussed as below: - Which parameters are the important to control WUE in this experiment? - How do these parameters affect WUE? - Are the estimated parameters reasonable compared to the previous studies?If not, why?

The latter paragraph concerns the plant water use efficiency (WUE) during the drought. We do not calibrate the model with WUE, but with ET and GPP. Hence the results concerning WUE cannot precede the discussion on parameter values or of the ET and GPP fluxes.

1-21. P18 Lines 22-23: I do not distinguish "the actual drought" in Fig. 3. I think it is better to add the period of the drought in the Fig. 3.

This will be added.

1-22. P18 Lines 27-33: I do not understand this paragraph because I do not understand Fig. 4. Does the lower panel show USO results? What is "Medlyn"? I also do not understand how is the β-function for the observation calculated.

Unfortunately this image contains the surname of the first author of the USO paper. This will be fixed. The lower panel are the USO model results. For the observations the coloring is mentioned in the brackets (Bethy dry or Medlyn dry) and in the image text. We have used the same intensity as in the middle column, applied for the corresponding day.

1-23. P18 Lines 34- P19 Line 5: The authors did not show the experimental setting and result explicitly, therefore I do not understand the purpose of this experiment. What is the difference between this experiment and the parameter optimization in section 3.3?

Within these lines we explain, that we examined the ET and GPP cycles with all conductance models and all sites with the generally optimised parameter set (all sites simultaneously) and the dry period set (only Hyytiälä dry period optimisation). So the optimised parameters are the same, but we expand the examination to all sites (so we check the ET and GPP cycles of other sites with Hyytiälä dry period calibration). This paragraph

highlights, that in general, the site level optimisation is poor, when applied to other sites and compared to the more general optimisation. We will clarify this at the beginning of the paragraph.

1-24. P19 Lines 17-20: I do not understand these sentences.

The parameter q only affects the model output, when soil moisture is below the fraction  $\theta$  tsp. Because this fraction was lowered, q is practically ineffective (this relates to comment 1-15). The dry period optimisation raised the fraction  $\theta$  tsp, so q was again effective. We will amend the sentences.

2-1. P17 Lines 7-13: The authors explain that the setting for USO is not appropriate. Then the results should not be used for further discussion after this paragraph. If authors would like to use the result, they should perform the experiment again with the appropriate settings.

We disagree with this comment, as we have explained above when answering the general comment concerning this topic.

**Technical corrections:**

1. P1 L11: "correctly time and replicate" -> "correctly reproduce"

Changed.

2. P2 Line 27: Abbreviation "APIS" should be placed here (this is the first appearance).

**Added.**

3. P9 Line 5: "from 2006" -> "in 2006" (only one-year optimization).

This is an incorrect correction, but we restructured the sentence.

4. P10 Lines 17-18: "high", "average", or "low" effectiveness value: this explanation should be the same as Table 4.Added "change in the parameter values".

**Modified.**

5. P11 Lines 19-20: Description of the Supplemental materials is needed at under each figure.

These will be added.

6. P15 Table5: some values are different form the Fig. 2 (i.e., r2 of Bethy).

We checked and noticed, that this is indeed so. The values in the images are automatically calculated, so they are correct. These will be corrected.

7. P18 Line13: "disregardin" -> "disregarding"

Corrected.

8. P17 Line 14 "The most noteworthy" what? (modified word is needed)

Added "change in the parameter values".

**Comments by Reviewer 2**

Dear editors, dear reviewers, I enjoyed reading this study by Mäkelä on the calibration of a new version of the JSBACH model. I find that topic and general approach fit well to Geoscientific Model Development, and that the paper has the potential to make an informative and useful contribution in this field. That being said, I currently see two major problems with the study (detailed below), aswell as a number of smaller issues that need to be cleared up before publication. The current abstract, and much of the method section, are concerned with the calibration of the model. At the same time, however, the authors make several modifications of the model (which are mostly described in the appendix), apparently in response to shortcomings that were identified in earlier studies, and present an additional case study (summer droughts in Hyytiälä) to demonstrate the improved properties of the model. As a reader, one gets the feeling that at least two studies were combined in one: i) a study about the calibration of JSBACH, with side notes about the effectiveness of the APIS algorithm ii) a study about model improvements. The case study on Hyytiälä seems to me, with due respect, a bit of a Finnish obsession – many models have problems with properly reproducing flux characteristics of Hyytiälä, I guess due the somewhat unusual soil / climate combination of this site, and although it's nice that the improved model fares better than the previous version, I'm not sure if there is a scientific reason for giving so much space to the performance of this site for a general vegetation model. To address this entire point, I would urge the authors to consider my comments, and think about whether this paper could / should be restructured, possibly by giving more space to model improvements in intro / methods.

One of the main reasons, why we have focused on the boreal evergreen forests, is to better understand the model deficiencies in replicating drought conditions. The stomatal conductance models formulate the plant response to these conditions. Because the models differ from one another, it is also important to calibrate the parameters accordingly. Therefore it seemed redundant to separate these aspects into two different papers and more beneficial to compare the model performance under drought conditions to the more general setting.

We understand the criticism towards the case study, but disagree with the implication. Contrary to the reviewers view, we have the experience that most models can replicate the Hyytiälä site flux characteristics quite well. The "case study" was made at Hyytiälä, because of the exceptional drought of 2006. Regionally it resulted in visible discoloration of needles, vegetation dying etc. that have been reported in Muukkonen et al (2015). Furthermore, the drought is visible in the Hyytiälä eddy covariance data in a manner we have no known record from any other Boreal coniferous EC site. The reason for examining the drought event, is because general vegetation models tend to run into problems when replicating droughts. These events are important to examine for the benefit of model improvement. It has also been speculated, that these events are likely to increase in frequency in the future. However, we will give more emphasis on general model improvements.

• The calibration procedure has several severe technical shortcomings that should be resolved. The most important is the formulation of the likelihood, which, at the moment, is not a real likelihood, but just an arbitrary cost function. You could also keep it like that, but then you shouldn't call the result posterior (as it is not based on a reasonable estimate of p(D|M)). Moreover, you should provide convergence diagnostics. If you want to make strong statements about the quality of APIS, I would recommend a benchmark against a suitable algorithm for you problem (I make a recommendation later). Moreover, I did not understand why an optimization is necessary on top of the posterior estimation. The MAP could also be estimated from the posterior sample(unless high precision about the exact location of the MAP is required)

This comment has a lot to do with the actual formulation of APIS and how it works, which apparently has not been conveyed well enough in the paper (as noted at the start of this document). We will clarify the descriptions. We do agree on the interpretation, as we have systematically called the objective function a "cost function". The word likelihood only appears when we are referring to the general Bayesian framework. We will reformulate the wording regarding the posterior pdf's. We will also add convergence diagnostics of the APIS parameter expected values. Benchmarking the algorithm against other algorithms is, however,

beyond the scope of this paper. The reasons for the use of an optimiser has been presented at the start of the document.

• Again, a bit of a broad comment, but I found that many of the conclusions are only weakly supported by the results, and also in the results and discussions, there are of points of interpretation that seem only weakly linked to the results. Could you please make sure, throughout the manuscript, that the discussion concentrates on tangible, numerical results, and that it is clear to the reader on what results you base your interpretation (e.g. by appropriate references to tables, figures, SI)

We do so in the revised manuscript.

**Detailed Comments:**

Title: says nothing about model improvements

1. 1.3 Name algorithm

We have added the abbreviation here. The modification (shape parameter adaptation) itself is not enough to warrant a new name for the algorithm.

2. 1.7 this sounds very vague – how was performance compared, and why do you say on the one hand that there was no clear best model, and then that some models were better.

This relates to both the cost function values (of optimised parameter sets), but also to site specific correlation and bias. The differences are too small to make definite statements about the best stomatal conductance formulation, but the individual metrics indicate better performance for some. We will improve the abstract in this regard.

3. 1.8 why would the improvement in the Finnish site be important?

The improvements are not important, because the site is in Finland, but due to the specific drought conditions we are interested in.

4. 1.10 This seems a completely unconnected question that is suddenly introduced here at the end of the abstract. See main comment 1

Yes this does seem to be a disconnected addition and we will improve the motivation.

5. 2.12 logical gap here – not clear why the problems named before call for species / zone specific parameterizations.

The drought responses are (usually) extensively generalised processes utilising bulk parameters. Soil moisture drought and vapour pressure deficit (VPD) affect different plants differently under various environmental conditions. In this paper we examine the drought effect on one plant functional type (PFT, not a species). It is much easier to examine the effects without the added complications of many PFTs.

6. 2.19 Why can this be hypothesized? To me, the only logical reason is that all models have (different) model errors, which is thus always compensated (differently) by other parts of the model. Is that your logic? If so, please make this explicit

This is broadly the same argument as ours and we will make this more explicit.

7. 2.25 It does not become clear why it would be necessary to study inter-site variability or the specific drought even for the questions you have posed before

Both of these questions are important, when we consider model caveats and deficiencies. Especially the droughts represent conditions, where many land-surface or ecosystem models fail to correctly replicate the observations.

8. 2.29 This seems to contradict the abstract, where you state you use an optimizer for the optimum. But you could of course have estimated the mode via a suitable density estimator, or just take the parameter value with highest posterior within the sampled points.

The end state of the APIS algorithm reflects all the modes of the target, so there is no contradiction here. The latter remarks are true but could not be directly used with some of the simulations (as explained at the start of the document).

9. 2.30 I'm not sure why you provide this information at this point – an overview about the methods would have been more logical

Will be moved.

10. 3.3 by aggregates you mean means?

Usually yes, but this depends on the units of the variable. In some instances, rainfall can be given in just mm without specified time (so in this case the values would be summed).

11. 3.8 The temporal split is of course a less independent validation than a new site, but OK, why not...

The independent validation sites are always valuable, but in many cases we are dealing with a limited amount of available data. In these cases it is usually more beneficial to split the timeseries into separate sections.

12. 4.3 What do you mean by "observational meteorological dataset" – where the weather stations on the flux sites? If not, how far away, and is that a problem?

All of the measurements are taken at the sites themselves (not including the specific data we mention in the manuscript). By meteorological we mean that these are site level measurements of the (current) meteorological conditions. Usually eddy flux sites have instrumentation that is at least on the level of typical weather station.

13. 4.5 I'm not really sure why you would want to consider a feedback in this context, i.e. if you have climate measurements on site. Probably relates to previous question

We are pointing here to model deficiencies, which may affect the results. This is a known problem with JSBACH (uncoupled) simulations especially when run with prescribed meteorological data. In certain winter conditions the model tries to balance the energy flux by condensing water. Additionally "climate" and "meteorological "measurements" are different things – climate is typically a statistical value (sum, mean or deviation) of meteorological conditions of a longer time period e.g. 30 years.

14. 4.8 Why do you need two citations for that fact that you don't work on a grid, but plot-based, i.e. for what are those references cited?

This contrasts more to the general difference of running the model on a grid (and what problems we get there) and a site level simulation. Some other papers require these types of distinctions. We will likely remove the references and simply state that the simulations are done on site-level.

15. 4.15 "fractional structure"? I think the first part of the sentence is just clutter, just say: In JSBACH, the land surface is divided into grid cells, and the grid cells are divided intotiles...

Agreed.

16. 4.17 site-level

**Corrected.**

17. 4.21 Although this seems logical on the first glance, it's not always clear if the "right" LAI setting for a model is the measured LAI, because models often assumes homogenous leaf distributions, but real leaf distribution is inhomogeneous, a lower-than-measured LAI will sometimes produce more appropriate photosynthesis values (cf Medlyn, Belinda E. "Physiological basis of the light use efficiency model." Tree physiology 18.3(1998): 167-176.). It depends on the model structure. I wonder if this would better be calibrated as well, or at least I'd like to hear your comments about the assumptions about leaf distribution in JSBACH and if setting observed LAI is clearly appropriate.

The "right" LAI is a tricky question that also depends on how the model is applied. We are considering boreal coniferous forests, where light penetration is deep. The light conditions themselves are more homogenous than for deciduous trees and therefore we can also assume more homogenous leaf distribution. JSBACH also takes into account leaf clumping. One of the more difficult aspects is the leaf shape (cylindrical) and orientation – these we assume similar throughout the study sites. We will consider adding some discussion about this to the paper. Optimisation, focusing on radiative effects, might be interesting, although JSBACH only utilises the two-stream approach.

18. 5.2 You give no reasons, but I assume the modifications were done to facilitate the calibration?

Yes, otherwise the model would have to be recompiled each time a parameter values is changed.

19. 5.7 Why give numbers for the groups and not a name?

Group names would take more space in the tables.

5.5. The sentence is unintelligible. Moreover, the explanation of how parameter ranges (i.e. priors
 – why don't you call them priors) are derived is not sufficient. Provide a clear rationale for prior
 elicitation.

Parameter ranges are not priors. This is explained at the beginning of this document.

21. 6.1 "The" lacking. In general, you are very economic regarding the use of articles.

Added.

22. 6.3 heatsum sum

Corrected.

23. 6.6 ready

We have modified the sentence as per request if the other reviewer.

24. 7.11 You didn't define Chi, but I assume this is your prior space? Also, there is no need that this space is a subset of Rn (you can have discrete parameters)

Yes. We will be falling back to the standard notation.

25. 7.11 Likewise, observations don't have to be continuous, thus not element R

As above.

26. 6.15 What do you mean by directly assessed?

This statement was made to reflect that we do not a have a readily available distribution that would also correctly reflect different observational sets (and more widely, different PFTs, soil conditions etc.). We will amend the sentence.

27. eq 3: the sense of the three different formulations of the right side of the formula evades me. The middle one is Bayes formula, the other two seem nonsense. If you want to define p(x|y) as l(x|y), why not define this directly. Moreover, usual notation for Likelihood is curly capital L. Same for g(x) – why first introduce the prior as p(x) and then rename it to g(x)? I also see no need for Z – if we keep on writing p(y), eq. 4 is much easier to understand

The reason was stated in the sentence, following this equation, where we noted that we are following the notation of Martino et al. (2015). This was done, so it would be easier for the reader to refer to this paper. We will be reverting back to the standard notation.

28. 7.25 It is a VERY unorthodox notation to define pi(x) as the posterior pi(x) is often used for the prior, to distinguish it from p(x|D). I found this highly confusing

Will be amended.

29. eq. 4 This seems to me a crazy reformulation of the formula, as it is so much harder to see why this holds as if you would just write the standard p(D) = int p(D | x) p(x) dx, which shows that if you marginalize the posterior over the space X, you are left with p(D).

**Agreed.**

30. 8.30 This entire procedure remained nebulous to me. First of all, if you favor the proposal mean, you should correct this in the acceptance probabilities, right?. How was this done? Secondly, when I understand correctly, you use the same spin-up (from the mean) for all parameters? I don't see how this can be justified, and how this could be corrected. What does "slightly scale" mean, do you increase or decrease weights?

APIS uses blind adaptation, so there are no acceptance probabilities. Each time the IS estimators are adapted, we generate new spin-ups (for each IS using the location of that estimator). We use the locations as the first draw, since the spin-up was generated with these parameter values and in APIS we are mostly interested in the proposal means. This means that we do favour the mean (as 1 of 50 draws is predetermined). In practice, we are (slightly) diminishing the rate of convergence (how "far" is the next IS location).

The purpose of the spin-up is to define the initial state for the model, and it mainly affects the "reservoirs" of soil water content and LAI. The difference in using the "correct" spin-up and one generated with the mean value, is typically small (cost function values within 1% of one another). This estimate is based on previous work, where we have dealt with the issue as well. The scaling (decrease of weights) is used to reflect our confidence in the draws. This procedure, and the one described in the paragraph above, both mainly affect the adaptation of the location parameters (the scaling is not used in the global estimates).

31. 9.9 If I understand correctly, you developed a new optimizer here? Why not use a well-known, tested optimization algorithm? In general, what you do here looks like a pretty standard gradient descent method. I would suggest to re-run this with an established optimizer (apart from the fact that I don't understand why you need an optimizer)

The method itself is more closely related to HMC as we are not estimating the gradient (or even trying to). We did not want to revert to the more usual gradient based methods, since we (somewhat) criticize these in the paper and it would have been a bit hypocritical. If it is required, we can test the stability of the optimisation (starting from the optimised values) with some common algorithm. The dislike of a method itself, is not a good enough reason to rerun all of the optimisations.

32. 9.20 I have many doubts whether this algorithm makes sense / performs better than alternatives, and would recommend to test optima against a reliable algorithm (DEoptim package in R is very reliable for complicated target in my experience), but OK, it's probably not the main point about

this paper. I just don't understand why you wouldn't fall back on standard solutions wherever possible.

This question closely relates to the one above, as does the answer. One of the reasons to use new methods, is to test how they perform. In this sense, it is not the most important thing if they are the best in the field, rather if they can get the job done.

33. 10.4 The spin-up procedure seems to favor the proposal mean and could thus distort the posterior. Please discuss

This was discussed above in the answer to 30.

34. 10.9 I'm not sure if I understand correctly – you are applying a KDE on the sampled posterior, and then create samples from that for the posterior predictive distribution? Why would you do that? Would that (potentially) distort the posterior? Please discuss and if you do what I think you do, prove that this does not distort the posterior.

The KDE is used purely for visualization and we do not draw samples from this. We use KDE to estimate the distribution behind the location parameters (snapshots at specified iterations) of the IS samplers.

35. 10.13 What's the sense of this effectiveness? It seems this is something like sensitivity, but this could be calculated directly from the difference prior – posterior. Moreover, when I understand correctly, this is a kind of conditional calculation, where you keep the other parameters at the optimum? Makes kind of sense, but is also loosing info about the parameter correlations in the posterior, so in theory, a parameter could be very "effective", despite being globally poorly constrained (due to a trade-off with another parameter). Please discuss.

The reason to add this "effectiveness" to the paper was to give the reader a sense of which parameters affect most the situation in JSBACH/APIS. It was not meant to be an exact measurement of anything (and we do not present these values in the manuscript). The equifinality of (two hypothetical) parameters is indeed something this measure would not capture (or rather it would capture the "effectiveness" but it would not be constrained as the reviewer suggests). For this we do not have a better answer other than to state that we do not seem to have encountered this type of situation. Our previous work on the subject (Makela et al. 2016) did not reveal considerable correlations (linear or otherwise) between the parameters and there is no indications that the processes (that are controlled by these parameters) themselves would support this.

36. 10.25 Based on your exposition, you should define a likelihood, and not a cost function. This word has no meaning in a Bayesian context.

We will discuss this in detail when answering the comment 40.

37. 10.25 If this is a likelihood, correct interpretation would be that likelihood is normal

Yes if indeed it is interpreted as a likelihood, it would be normal.

38. 10.26 a) what do you mean by "successfully done" – that it went through the review? please give a reason for this b) so, the cost function is NOT the MSE

Yes, this should be more in the lines that this type of cost function has been used before in similar settings. We will remove the word "successfully". We stated that the cost function is "based on the mean squared error" as it is indeed not the MSE. The terminology used here will be revised.

39. 10.29 Not clear to me what you mean by "covariance vector", and "combining model and observation error". You don't know the model error as such, but of course, as in a linear regression, you can fit the sd of the normal distribution to the effective spread around the predicted value, which is the standard approach.

This comment in the manuscript was a reflection to the standard Bayesian framework, where both error terms are used.

40. eq 9 – OK, if you want to define this as your likelihood, then you should simply state the correct assumptions. What you assume here is that the relative error is normally distributed, with a standard deviation EQUAL the mean, and additionally you divide the likelihood by the number of data points (dividing by N\_ET), which makes no sense if you truly want to create a likelihood. Why does this not make sense?

a) you don't know a priori how your residual scatter around the model predictions. The scale of the normal distribution (essentially the denominator in the likelihood) affects the shape of the posterior, i.e. makes it more or less wide. As you can't know the correct scale, you have to fit a parameter here

b) also, it doesn't make sense to have residual go to zero for small observations. A sensible expression for the scale of the normal would be to fit log likelihood = (predicted – observed / (a0 + a1 \*predicted))2where parameters a0, a1 have to be optimized.

c) there is no good statistical reason to divide by N\_ET, i.e. by using a mean squared error as the likelihood. Essentially, by doing this, you scale the likelihood to have the evidence of one data point, making the posterior much wider than it would naturally be In general, it seems to me that the likelihood you use here creates a posterior that is far wider than any sensible statistical assumptions would allow.

These are all valid remarks and we will state the assumptions explicitly in the revised paper. As stated before, APIS works by estimating the expected value of the distribution. As per design, it works very much in the sense of a MAP estimate (which is unaffected by the scaling). The scaling (dividing by the number of unmasked data points) does indeed inflate/deflate the distribution, which is accounted for by the adaptation of the shape parameters in the APIS. The reason to include this divider, is that otherwise the cost function would be biased towards certain study sites. This formulations allowed us also to compare the single-site values and multi-site values directly etc.

Another remark we would like to make, is that we also tested a likelihood (although on a single site and I can't seem to find the results/chains) with the observational error of 20% of the flux value (and using the same scaling). We used the wintertime observations to estimate the precision for each variable. The difference in these results was small. We also wanted to be able to compare the results directly to Knauer et al. (2015) that contained previous work on the subject with JSBACH and utilized the same formulations. Hence, the benefits of translating (in this work) to the new formulation, seemed small when we considered the drawbacks.

The residuals that go to zero are of less importance in this type of work. Almost all of the near-zero values occur in wintertime which the model has some problems and we were initially debating whether to just mask all wintertime values. Additionally, on experience the model results in wintertime are near identical (this holds especially for GPP that should be zero in winter, the ET suffers from e.g. the lack of closure of the energy balance mentioned in the manuscript).

41. 11.8 I found the structure with results and discussion together not very helpful. It seems to me that this is adding to the fragmentation of this paper, which seems to address several questions (model improvement, calibration, drought case study) at the same time. A discussion which summarizes the results and puts them in a common perspective would have seemed preferable to me

Thank you for the comment, we will be separating these into two sections.

42. 11.13 it seems you suggest in this paragraph that identifiability equals or is related to convergence, and it's not clear to me why (in general, these are two different issues). Moreover, I can see no visual difference in convergence speed between the three examples. If you claim the first one converges faster, please back this up by numeric estimates of convergence, e.g. Gelman-Rubin.

We will be adding the convergence test to the paper.

43. 11.13 Moreover, you should provide convergence diagnostics / proof of convergence (typically Gelman-Rubin) for all your results! Not having checked convergence is not acceptable.

These are readily available. However, convergence can not be proven. What can be shown, is the lack of divergence.

44. 11.16 It's not the algorithm that is unable to constrain the parameter, it's the likelihood. Also, you seem to suggest that this is a problem, but that's perfectly normal for a Bayesian analysis.

In the sense, that the probability distribution is defined by the likelihood, this is absolutely true. Here we were referring more to the fact that in APIS, the draws are not evaluated one-by-one, but in groups of 50, so the characteristics of one parameter can be masked by the characteristics of another.

45. 11.20 What does reasonably stable mean? See comment about convergence diagnostics above

We will be adding the convergence test results.

46. 11.24 Again, not really clear to me why you do the optimization in the first place, instead of estimating the MAP from the posterior sample

Answered at the start of this document.

47. 11.25 Near or at. Why is near the limits a problem? If you have flat priors, you state that all these values are equally likely, so near limit is no problem. I suspect though that you have MAPs at the limit, posterior medians are of course never at the limit.

There're both cases, but this is mostly in reference to the relative humidity fraction (which is at the limit).

48. 12.8 You can parallelize the chains in DREAM, which means that, assuming you run MCMCs as usual, you can use at least 9 cores. I'm not sure how many cores you were using. On the plus side, I'd bet that DREAM or DEzs algorithms converge faster than APIS. I think it would be useful to benchmark against one of these algorithms. Both are implemented in the R package BayesianTools.

This characteristics of DREAM is mentioned in the next lines. In our setting for APIS, the two thousand (2000) draws (50 draws from 40 IS samplers) can be estimated and run simultaneously (we did not run it this way, as the resources are limited). The main difference here, is that APIS requires only a fraction of the amount of sequential draws (this is the point we are making). In this paper, we have demonstrated, that APIS can be used in these types of situations. It is clearly not the best in the case of unimodal (or even with few peaks) distributions, but it should be kept in mind when estimating more complicated targets.

49. 12.20 So, why not calibrate them right away?

The original idea was to restrict the calibration to g0 and g1 only so that all of the Ball-Berry variants would be on the same "line".

50. 12.20 / 12.26 Most of the info in these paragraphs are not results

Will be moved.

51. 16.22 The entire section reads like an independent case study with its own methods, results and discussion.

This will likely change as we separate the "Results and discussion" into two sections.

52. 19.8 Comments about APIS: I could not see a serious evaluation of the convergence and quality of this algorithm in the paper. At least, you should provide convergence checks. If you want to say anything about the quality of APIS, I think you should compare to a reasonable reference. For example, DEzs or DREAMzs in the R package BayesianTools would be suitable reference algorithms that have proven to work well for these kinds of problems.

The convergence test results will be added to the paper. The comparison to other algorithms is beyond the scope of this paper.

53. 19.10 Define successful.

We will amend the sentence and make this more explicit.

54. 19.10 General comment: for any claim you make in this section about your findings, please refer to a specific result in section 3 that is the basis of your claim. Specifically, I can see no results that provide hard support for your first two claims.

We will divide the "Results and discussion" into two sections and add the convergence tests.

55. 19.28 Code and data availability is insufficient. Unless there is a good reason against this, please provide all code and empirical data (drivers and calibration / validation) with the paper, or in an appropriate repository. FLUXNET could be updated or changed, and in any case, it would be more convenient for the reader to have your entire data set at hand. Moreover, you should ensure computational reproducibility, but storing random seeds etc. for the algorithms. Ideally, also results, in particular MCMC chains should be saved, if space permits this.

The code is under MPI-M License agreement and we cannot distribute it. The driving data (approximately 500Mb) and chains can be uploaded e.g. as supplements.

---

## Author Response (AR1)

**GMD-2018-313: Stomatal conductance, photosynthesis and parameter calibration for boreal forests with adaptive population importance sampler in the land surface model JSBACH**

This document contains the marked-up differences between the previous version of the manuscript and the revised version of gmd-2018-313. Please note that we had to remove the tables and some of the equations from the previous version because latexdiff was not able to process the changes (probably this was because of the multicolumns in Tables etc.). Additionally, latexdiff places most of the new tables at the end of the manuscript.

We have already previously uploaded the point-by-point answers to the reviewer comments as a supplement to the author comment AC1: "Author responses to the reviewer and editor comments" (Jarmo Mäkelä, 02 May 2019).

The revised manuscript has been modified according to the comments and our answers. The major modifications are:

- Revised title.

- More focus on APIS in the introduction.

- Revised overview on the model simulations with added details.

- Revised text of the APIS algorithm description with comparison to a general MCMC method.

- Added Gelman-Rubin tests.

- Separated and revised the "Results and discussion" into two separate section. The text has been mostly restructured but there are also some new points made.

- The new discussion focuses on the validity of simulations and the model modifications, with some further comments on the parameter values.

- Figures have been modified and captions added to the supplementary images.

- The previously wrong numbers of $r^2$ in Table 5 (now Table 6) have been corrected (the $r^2$ values in the table had not been squared).

- The algorithm states have been added as supplements.

- The driving data has been uploaded to Zenodo.

- The model modifications have been uploaded to github but access still requires agreement to the MPI-M License agreement.

[revised manuscript text omitted]

| CA-Qfo | 49.69 | -74.34 | 382 | *Picea mariana* | 3.7 | 112 | 962 | -0.4 | 2003–2010 | **?** |
| FI-Hyy | 61.85 | 24.29 | 180 | *Pinus sylvestris* | 3.5 | 45 | 709 | 2.9 | 1999–2006 | **?** |
| FI-Ken | 67.99 | 24.24 | 337 | *Picea abies* | 2.1 | 100 | 484 | 0.4 | 2003–2010 | **?** |
| FI-Sod | 67.36 | 26.64 | 179 | *Pinus sylvestris* | 1.7 | 150 | 527 | -0.4 | 2001–2008 | **?** |
| RU-Fyo | 56.45 | 32.90 | 265 | *Picea abies* | 4.5 | 200 | 711 | 3.9 | 2002–2009 | **?** |
| CA-Ojp | 53.92 | -104.69 | 579 | *Pinus banksiana* | 2.6 | 100 | 431 | 0.1 | 2004–2006 | **?** |
| FI-Let | 60.64 | 23.96 | 119 | *Pinus sylvestris* | 6.0 | 40 | 627 | 4.6 | 2010–2012 | **?** |
| RU-Zot | 60.80 | 89.35 | 121 | *Pinus sylvestris* | 1.5 | 215 | 493 | -3.3 | 2002–2004 | **?** |
| US-Prr | 65.12 | -147.49 | 210 | *Picea mariana* | 0.7 | 72 | 275 | -2.0 | 2011–2013 | **?** |

**2.2 The JSBACH model**

[revised manuscript text omitted]

---

## Referee Report (RR1)

Dear authors,

I checked the authors appropriately addressed all items raised by the referee #2 except following 2 items.

(1) For the following referee's comment.

> 20. 5.5. The sentence is unintelligible. Moreover, the explanation of how parameter ranges (i.e. priors - why don't you call them priors) are derived is not sufficient. Provide a clear rationale for prior elicitation.

Authors replied as follows.

> Parameter ranges are not priors. This is explained at the beginning of this document.

But I could not find corresponding explanation on the manuscript.

(2) On the last item from the referee#2, authors replied as follows.

> The code is under MPI-M License agreement and we cannot distribute it. The driving data (approximately 500Mb) and chains can be uploaded e.g. as supplements.

For ensuring computational reproducibility, driving data, at least, should be available on an appropriate repository, which is reasonably accessible for readers.

Best,
Hisashi SATO

---

## Author Response (AR2)

This document contains the reviewer comments (and our responses) to the second iteration of the manuscript: https://www.geosci-model-dev-discuss.net/gmd-2018-313/

- attached is also a new supplement S3 (see response to general comment 2)
- and the latexdiff file containing the modifications to the previous version of the manuscript (the images have been cut from the file to keep the file size moderate).

The comments are gathered here as indented text and we have added numbering to them, so they can be individually referred to, if need be. Our responses to these comments are shown as unindented text (such as this) after each comment. We have made few additional grammar corrections, and also modified Fig. 4 by changing the axis units (so both are in mass units) and added gridlines to the image.

**Comments by Reviewer 1**

I thank the authors for correcting the previous manuscript according to my comments. The procedure of the experiment becomes much clearer, and generally, I agree with the interpretation of the results. However, now I found some contradictions in the results and discussion. Authors need to carefully explain the results for ET and the estimated parameters related to the water stress. As the other reviewer suggested, the conclusions should be supported by the numerical results.

I also suggest some improvements for the manuscript so that the readers can easily understand (notified with "Draft amendment"). They are just my recommendations for the improvements of the manuscript, and there may be misunderstandings. Therefore, the authors needed to reconsider the way of writings. I read these sentences, again and again, therefore, the difficulties should be the same for other readers.

**General comments:**

The authors need to describe the below results carefully.

1. Improvement of springtime increase

   GPP is greatly improved indeed, but improvement for ET is not so clear.

On this we agree but it should also be noted that there is not such an obvious discrepancy between the modelled and observed springtime ET as there is for GPP. We can see improvements e.g in FI-Hyy and FI-Sod, where the default model early bias has been removed.

2. b and r^2

   ○ Fig. 2 and S2 show that both b and r^2 are improved for GPP. However, b for ET is not so much improved or sometimes gets worse (especially the biases at FI-Ken and FL-Sod are apparent). I think it is better to show the results which get worse compared to the default using italic letters in Table 6. Then the bias increment for ET at FI-Ken, FL-Sod, and both for ET and GPP at RU-Zot and US-Prr will also be clear.

The focus of Table 6 is to identify the "best" conductance formulation from the results point of view whereas the suggested metric would show differences between default and optimised model versions and thus evaluate the performance of the calibration. We believe that combining the suggested metric to this table would not be beneficial as it would blur the focus that is to compare the conductance model behaviour. However, we have produced a bit more detailed analysis along the suggested lines and added it as a supporting supplement S3. The supplement contains details on both model specific and site specific improvements and deterioration's in the calibration process, separated for GPP and ET and the metrics (b and r²). The bias increments and deteriorations in the model are clearly apparent in the supplement.

We are treating these results only as supporting material and not including them in the main text because the interpretation of the analysis is not straightforward. In the supplement we are not comparing like-to-like – the JSBACH default parameters reflect the Baseline and Bethy behaviour (as they are originally part of the model) but the BB (and variants) are newly introduced to the JSBACH. The default parameters for the BB models can be seen as arbitrary and we have not done/presented any initial calibration/validation. Therefore, we like to keep the focus of the manuscript in the comparison of the "end results".

- ○ r^2 for ET is improved compared to the default, but much lower than r^2 for GPP. These results are also needed to be explained in the result section. The bold letter may be better for, e.g. from 0.9 to 1.1 so that the readers easily recognize the higher values.

The main reasons for this is that in JSBACH, e.g. the conductance is resolved for carbon assimilation, and the same conductance is then used for transpiration. In the model, GPP always takes a priority and is the determining factor (we could almost say that ET/transpiration is an afterthought). Additionally, GPP is derived from EC measurements by flux partitioning – this tends remove some of the flux instabilities (that are still present in ET). This also plays a role as the more stable GPP flux is easier to model than the more chaotic ET. This explanation was added to discussion in section "Validity of the simulations".

We appreciate the suggestion about the metric and explored the idea but ultimately feel that our original "metric" is more suited for the situation. The purpose of the bolded values is not to draw attention to where the model reproduces the observations well. The focus is on the comparison of the conductance formulations and systematic differences in the model performance – so we are also interested in the best performance on poorly replicated sites. The suggested metric would only focus on the "good" sites.

In addition to the suggestion, we also tested highlighting the best value (for each site and b, r² separately) and all values that are within 5% of it (similarly to S3). This approach yielded similar, but slightly obscured results to the original metric (so the interpretation was not as straightforward).

- ○ For model validation, b and r^2 may better to be evaluated separately to prevent the overestimate, because sometimes r^2 is high but b gets worth. From this aspect, it may better to cumulate the number of the "good" result e.g. from 0.9 to 1.1 for b and 0.9 to 1.0 for r^2, respectively. Then b and r^2 is better to be explained and discussed for ET and GPP respectively.

We do not follow, what overestimate the reviewer means. The question about using a different metric to examine the model performance was addressed above. We have separated the ET and the GPP discussion along these lines, in the places where the reviewer has suggested it (see the specific comments).

The "good" results yielded with the suggested metric (with the suggested limits) are presented below and they do not provide clear added value to the results of Table 6:

|        | Base | Bethy | BB | Leu | F&K | USO |
|--------|------|-------|----|-----|-----|-----|
| ET: b  | 5    | 5     | 5  | 4   | 3   | 1   |
| ET: r² | 0    | 0     | 0  | 0   | 0   | 0   |
| GPP: b | 3    | 4     | 5  | 4   | 4   | 4   |
| GPP: r²| 3    | 3     | 2  | 4   | 5   | 5   |

3. q, θpwp, θtsp

The effects of these parameters are complicated. Therefore, a clear explanation is needed so that the readers can understand it easily. The optimised θpwp and θtsp for the general

condition are smaller than the default; that makes β larger (which means, β became ineffective). The oprimised θ tsp for the dry condition is also smaller than the default but larger than that of general condition and θpwp varies between the sites; that makes β different between the sites. Larger q makes β smaller, whereas smaller q makes β larger (because β is between 0 and 1). In my understanding, q changes An, which control GPP but not affect ET. Is that correct?

We have added description about the combined effect of these parameters to the end of the section 2.3 "Modifications to the JSBACH model". The changes to parameter values do not necessarily make β ineffective, they can also be viewed to change the response intensity to drought (instead of slow gradual restriction we get a more abrupt and strong response). The parameter θpwp does not vary between the sites in our simulations – it varies between the conductance formulations (i.e. Fig. 3 rightmost panels) as the reviewer probably meant.

The β function is first used to calculate the stomatal conductance for carbon  assimilation. The same conductance is later used for transpiration calculations (A8). However, the factor $β^q$ only affects the net assimilation, not ET. So q affects only GPP.

> 4. I understand that the authors used the mean of the site level cost function values for APIS. Then, the parameters of the 50 draw for each IS sampler are the same for all study site? Does Figure 1 show the "global" estimate by using the "mean" of the site level cost function? I think it is better to add these descriptions to P9 Lines 25-27 and Fig. 1 so that the readers can easily understand the procedure.

Essentially, each site generates the ET and GPP fluxes with the same parameter values. Then the actual cost function (which is returned to APIS/optimiser) is calculated as the mean of the site level "cost functions". So when we refer to the cost function, it is always this mean (except for the dry period). We modified the cost function definition in 2.9 (which we believe is the correct location) to better reflect this.

All of the estimates in Fig. 1 are calculated with the "average cost function". The  global estimate (yellow) is produced with the deterministic mixture approach (item 4 in the MCMC-APIS comparison list) whereas the red is just the mean of the IS sampler locations. We added the mention of the deterministic mixture to the image caption.

**Specific Comments:**

> 1. P1. The title is needed to be reconsidered. For example, "Parameter calibration for stomatal conductance and photosynthesis.."

We calibrate also parameters that are not directly linked to stomatal conductance or photosynthesis, so the suggested revision is imprecise. AS a compromise, we could change the title to: "Parameter calibration and stomatal conductance formulation comparison for boreal forests with adaptive population importance sampler in the land surface model JSBACH"

> 2. P1. Lines 6-8: Draft amendment, also please reconsider the description with regarding general comments 1. "This modification enabled the model to correctly reproduce the springtime increase in GPP for conifers throughout the measurements sites used in this study. However, the improvement for ET was limited. The key parameters identified along with this modification were the parameters which control the soil moisture stress function and the overall rate of carbon fixation."

> 3. P1. Lines 8-10: Please reconsider the description, "Overall, … models", concerning general comments 2.

We have amended the manuscript to better reflect the two suggestions above. However, the delayed effect of temperature is mostly correcting the erroneous behaviour of the springtime GPP – the springtime ET is not the "original" problem, so it should not be the focus here. We now state the "improvements" for ET and GPP separately in the manuscript.

4. P5. Line21: What is "new"? Is that mean new to the JSBACH model? The explanation for q (one of the key parameters) is also not found in the main text.

Yes, the text has been amended to reflect this better. We have also added the explanations of q to the end of the section 2.3.

5. P10. Lines 4-11: I do not understand this paragraph. Please reconsider the descriptions. In my understanding, each IS sampler use just one parameter set for spin-up. Then the remaining 49 members use the same spin-up but the parameters are perturbed around the first parameter set. If so, such description may help the readers to understand the spin-up process easily. The initial parameter combination for each IS sampler is also selected randomly from the ranges in Table 2. This information is also needed. I also do not understand the procedure, "We also slightly scale (reduce) the importance weights based on the distance of ..". Is it the same as procedure 3 or different procedure just for spin-up?

We rearranged the sentenced in the paragraph and added the proposed description. The initial parameter combination is now more explicitly stated in the MCMC-APIS comparison list. The scaling procedure is the same as depicted in the list before.

6. P11. Lines 6-8: I still do not understand "Since we also did not run the model spin-up for .. parameter values". The authors described that the post procedure for APIS is used for the "correct spin-up" at P10. Lines 4-11. Is not enough? If so, why?

We do believe that the "spin-up correction" should be sufficient but it is also a good practice to verify the results of the more complicated sampling algorithms with simpler optimisers (as we have done here). The optimiser is also the only algorithm used for the drought period (we noticed this detail was missing from the manuscript and added it to this section).

7. P11. Line 8: Does "the same datasets as APIS" mean, same calibration period (the first five years), same climate forcing data, and the same observation data? Is the initial state also generated by APIS, or same as spin-up for APIS? Please make these settings clear.

The dataset is exactly the same as for APIS. The initial state is the mean of the APIS final location parameters. The spin-up is run separately for all the samples the optimiser draws. These details have been added to the manuscript.

8. P12. Line 5: What are the "acceptable values"? Is that mean the final state of the parameter distribution range of the parameter optimisation? Please make it clear.

No, this refers to the absolute range of the parameters. We added a reference to the Table 2, which should make this clear.

9. P14. Lines 6-8: The former sentence indicates that bud burst is not as critical, whereas the latter sentence indicates that the acclimation parameter dominates the phenology parameters. At first, I thought these sentences contradict, but later I understand the difference of these procedures. I think it is better to describe the different functions of these parameters clearly so that the readers easily understand.

We added a description for the function of tau and the LoGro parameters.

10. P15. Lines 10-11: "The annual cycles of the Bethy model are more in line with the Ball-Berry variants than those of the Baseline model (see supplements S2 for the yearly cycles of the

other models)." I think this is not always true. This result is not used in discussion, so I think it is better to remove this sentence. The authors can rewrite that "The results of other stomatal conductance models are shown in S2".

We modified the sentences according to the suggestions. We also added here a mention of the supporting material in supplement S3.

11. P15. Lines 15-18: Please reconsider this paragraph according to my general comment 1 and 2.

We modified this paragraph and stated separately the ET and GPP behaviour.

12. P16. Table 6: "best values" -> "N over threshold" (please refer general comment 2).

Not changed as we did not change the metric (see answer to general comment 2).

13. P17. Line 10-16: The authors do not show the results. Therefore, I do not understand the experimental setting and what is validated.

We produce similar results as in Fig. 2 and supplement S2 but using the dry period optimised parameters. We compare these to Fig. 2 and S2. The paragraph in question merely states that the dry period optimisation does not generally produce better results. We amended the paragraph so the setup is easier to understand.

14. P18. Lines 24-26: "This is mostly a direct result of the normalisation of the cost function that inflates the target distribution and gives too much weight to the initial locations and draws." I do not understand this sentence. I think the convergence is rather related to the parameter sensitivity to the observations.

Yes, the reviewer is absolutely correct. The cost function depicts the parameter sensitivity to observations. So when we normalise the cost function, we inflate the target distribution – so the parameter (relative) sensitivity to the observations is reduced.

15. P19. Lines 1-5: Draft amendment, also please refer general comment 1.

"This delay is also reflected in transpiration, and consequently in ET at FI-Hyy and FI-Sod to some extent. However, the effect at the other sites is not clear." (see general comment 1). The description for FI-Sod seems too much detailed, and discussion is rather needed to be done for general comment 2. Also, I wonder why ET is not improved greatly compared to GPP, although both the observations are used for the calibration. There may be some possible reasons for the mismatch. 1) interaction in the optimisaion process. For me, it seems that GPP is optimised by soil water parameters, and that affect ET estimate. 2) Parameter estimation bias (e.g. θhum strongly decreased and get to its lower limit). 3) Bias correction by q. In my understanding, GPP bias can be corrected using q, but q does not affect the bias of ET. Considering these issues are important for the study with multi observations and to improve ET. I recommend the authors to run some additional experiment to clarify the issues mentioned above for the discussion (only one site is enough). If it is beyond the scope, please discuss the possible reasons for future study.

We modified the sentences following the suggestions. We clarified the FI-Sod description and feel that it is an important point to make – improving model behaviour can lead to an increase in the ET bias if the improvement negates a previously erronous behaviour. The differences in ET/GPP improvements have been addressed in our response to general comment 2.

16. P19. Lines 13-16: Please reconsider this paragraph according to my general comment 2.

We modified this paragraph slightly, but as we did not change the metric in Table 6, the changes are minor.

17. P20. Lines 1-3: I do not find "the result of the site level estimates of g1", so I do not understand this sentence. "not only" is needed at Line 1. I also do not understand what does "control" for Wang mean.

This is a general comment regarding e.g. literature values, not specific simulations/results in the manuscript – the sentence was modified to reflect this. The control refers to the setting in the Wang (1996) paper, this has been modified to "(in Table 1, Control)", which should clarify the reference.

18. P20. Lines 18-28: Please reconsider this paragraph with general comments 3.

We added the clarifications in the "Model modifications" section as explained in the answer to general comment 3. Therefore, we mainly added some clarifications to this paragraph.

19. P20, Line 20: Not only $\theta_{tsp}$ but also $\theta_{pwp}$ is lower.

Added.

20. P21. Lines 3-4: "The parameters affecting the optimisation process the most were consistent for all stomatal conductance formulations." I do not understand this sentence.

Yes this sentence was missing commas, but we modified the sentence to be more understandable: "The parameters that were most effective in the optimisation processes, were consistent for all stomatal conductance formulations."

21. P21. Lines 8-11: Please reconsider this paragraph with general comments 3.

○ How did the authors evaluate the "importance of q for the Ball-Berry type model"? The same validation of importance in Table 5 may need for Table 7.

○ "Overall, both optimisations strongly indicate that boreal forest transpiration is not limited by soil moisture stress under normal conditions." In the discussion, the authors indicated two reasons (the other is the water retention capabilities of the soil). Also, ET is sometimes underestimated. Therefore, the authors can just indicate the possibility.

The importance can be verified from the sampling states directly, but it can also be seen from Fig. 3 rightmost panels, where we have plotted the β-function values during the drought. Since we are focusing on the dry event, the relative amount of values affected by q (i.e. low soil moisture) is considerable higher than under general conditions (so q becomes more important). The "importance metric" was not used here because it was meant to reflect the APIS simulation identifiability (so it would be detrimental to add it to Table 7, where APIS was not used).

We added a mention about the nonlinearity of the additional reduction ($\beta^q$), which also indicates that soil moisture stress is not the limiting factor during normal conditions. We removed  the word "strongly" but feel that this formulation should be sufficient as it is indicative. If ET is underestimated, then likely transpiration is also – so this is not a contradiction as such but points to other possible problems in the model.

22. P21. Line 13: Please reconsider this paragraph with general comments 2.

We examined the different metric proposed in GC2, but decided against this. Therefore, only minimal changes were made to this paragraph.

**Technical corrections**

1. P1. Lines 2-3: Draft amendment

"The parameter posterior distributions were generated by the adaptive population importance sampler (APIS), then the optimal values were estimated by a simple stochastic optimisation algorithm"

Accepted.

2. P1. Lines 3-5: Draft amendment

"Using the in-situ measurements of evapotranspiration (ET) and gross primary production (GPP), we calibrated three model parameter groups (\*\*, \*\*, and \*\*), and identified the key parameters. "

This is slightly inaccurate, because we don't calibrate the groups – APIS does not use the grouping and the optimiser also draws samples from the full parameter space. We modified the previous formulation to include the "in-situ observations".

3. P1. Lines 11-13: Draft amendment

"This optimisation improved the model behaviour, but the changes to the parameter values were significant except for the unified stomatal optimization model (USO). Interestingly, the USO model demonstrated the best performance during this event with only small changes to the parameter values."

Accepted with modifications: "This optimisation improved the model behaviour, but resulted in significant changes to the parameter values except for the unified stomatal optimisation model (USO). Interestingly, the USO model demonstrated the best performance during this event."

4. P2. Lines 20-22: Draft amendment

"It can be hypothesised that the choice of the stomatal conductance model affects the ecosystem model parameters broadly. Because the stomatal conductance formulations vary in their responses to the different conditions. However, a holistic assessment of the performance of the stomatal conductance models together with other parameters (e.g. photosynthesis parameters) has been missing."

Accepted with slight modifications.

5. P3. Lines 6-7: Draft amendment

"The APIS algorithm samples the full parameter space (as do MCMC methods) and it can treat a mixture of parameter prior distributions. Therefore, APIS can estimate complicated multidimensional probability distributions."

Accepted with slight modifications.

6. P3. Lines 11-14: Draft amendment

"First, we utilise APIS to sample the full parameter space with the different stomatal conductance formulations and to locate different modes of the target parameter distributions (peaks of high probability). Second, using the distributions generated by APIS as the prior distributions, the parameters are optimized using a simple stochastic optimisation method. Finally, we assess the inter-site variability and the robustness of the calibrated parameters together with different stomatal conductance formulations. Optimised parameters for a specific drought is also investigated and compared with the parameters for the general optimisation."

Accepted with slight modifications.

7. P3. Line 20: Draft amendment

"The site level half-hourly measurements of eddy covariance (EC)"

Accepted with modified word order.

8. P3. Lines 21-24: Draft amendment

> "The gap-filled and low quality (based on FLUXNET data quality flags) measurements were masked, and the daily aggregates (usually means) were accepted as part of the calibration process if at least 60% of values between 4:00 and 20:00 (i.e. daytime measurements) for that day were unmasked. The daily aggregated data (ET and GPP) were used for the calibration and the validation, whereas all of the half-hourly data were used as the climate forcing data (as explained in section 2.4)."

Accepted with slight modifications.

9. P5. Line 32: Draft amendment

> "However, coniferous evergreen trees do not shed all of their leaves for winter, and the original phenology model is not suitable for a boreal forest."

This is not entirely correct. The purpose of the phenology model is to determine when new leaves start to grow and the consequent growth rate. The state of acclimation that in our simulations corrects the early spring GPP, restricts the net assimilation rate but does not (directly) influence the phenology model. We believe that the phenology model itself is performing adequately.

10. P6. Table 2: The additional parameters for Friend and Kiang model is also needed to be included.

Added.

11. P7. Line 1: There are many "b" in this manuscript: photosynthetic acclimation, additional parameter for Friend and Kiang model, and the slope of the regression line. The authors should change them so that the readers can recognize these parameters are different.

Modifed, the F&K exponent is now d and the curvature in photosynthetic acclimation is k.

12. P7. Line 17: Draft amendment

> "In the original JSBACH formulation (i.e. the Baseline version),"

Accepted.

13. P9. Lines 5-8: Draft amendment

> "Above i is the elements with each IS sampler (described later). Generally, Eq. (4) cannot be analytically solved, hence it is usually estimated numerically. Commonly this is achieved by one of the many Markov chain Monte Carlo (MCMC) methods, but in this study, we apply the adaptive population importance sampler (APIS) defined by Martino et al. (2015). APIS is a Monte Carlo (MC) method that utilises a population of importance samplers (IS) to jointly estimate the target pdf ($p(\theta|x)$) and the normalising constant ($Z(x)$) by a deterministic mixture approach (Veach and Guibas, 1995; Owen and Yi, 2000), whereas the MCMC methods do not care about the value of Z. Importance sampling density $q(\theta)$ is also introduced in APIS algorism."

> Then P10. Lines 13-15 needed to be removed to here.

Accepted with slight modifications. The "i" refers to the i-th element in the (parameter) vector as is the standard notation (we added this information also). It is just calculated as the marginal integral over the whole parameter space.

14. P11. Line 6: Draft amendment

> "overshadows the calculations" -> I do not understand "overshadow". Appropriate word is needed.

This was reworded to "dominates the calculations".

15. P13. Line 17: I could not find Ball-Berry results in S1.

The supplement included the image "APIS-S1-posteriors.png" that includes both Bethy and BB posteriors at 20 iterations. Apparently I forgot to add the chains for the BB results when I reproduced the other images. Will be added to the supplement.

16. P13. Table 4: Draft amendment

"Parameter scale reduction ^R (at APIS iteration) and stability δ(threshold number of the iteration) estimates from the Bethy simulations."

Accepted.

17. P14. Lines 2-3: Draft amendment

"There is an overall agreement on the values of the most prevalent parameters (see the bold and the italic letters in Table 5 between the models."

Accepted, with "letters" changed to characters.

18. P14. Table 5: The order of the parameters is different from Table 2. I think that the same order is better to be understood. "b" also should be reconsidered.

Yes, the g0, g1, a, d (previously b) were moved to the end of the Table. We will change this to the same order as before.

19. P15. Lines 26-27: Draft amendment

"The values of the relative humidity parameter θhum, the residual stomatal conductance g0, and fC3 have remained nearly unchanged,"

The relative change in the values of $f_{C3}$ is quite large and the parameter is now at the lower limit. We added remarks on this (and the low values of g1) to the text.

20. P15. Lines 28-29: Draft amendment

"Noticeably the USO optimisation only changes the value of θtsp and q, and leaves the rest of the parameters almost untouched."

Accepted with slight modifications.

21. P15. Lines 30-33: Draft amendment

"For ET, the Baseline, Ball-Berry, and USO are greatly improved especially at the drought in summer 2006 when compared to more general optimisation, however too much drawdown was found for Bethy. The Baseline, Ball-Berry, Leuning, and to a lesser degree the Friend and Kiang formulations, now suffer from the too low ET values before the actual drought. GPP was greatly improved both for general and dry period optimisatons except for the drawdown for the Baseline and Bethy at the drought in summer 2006. Drawdown for USO is also clear but successfully reproduce the observed drawdown. The GPP of other formulations has remained roughly the same as with the more generally optimised parameter values. Overall, The Bethy model has a too strong drawdown for both ET and GPP during the drought."

We did not directly accept this amendment (as there is some repetition), but modified the paragraph in question to clarify it.

22. P17. Line 4: "Fig. 4, right"

We added "rightmost panels".

23. P18. Line 9: Draft amendment

"reproducing the fluxes for the validation sites with low LAI (i.e. RU-Zot and US-Prr)"

Accepted.

24. L18. Line 17: Draft amendment

"We optimised the model for individual (calibration) sites as well (not shown)."

Accepted.

25. P20. Line 1: "The site level estimates of (g0 and) g1 are sensitive not only to"

Accepted.

26. P3. Figure 4: "Bethy (general)" seems better.

We are not entirely certain, what the reviewer indicates here. In Fig. 4, we can understand why Bethy (opt/general) seems better than Bethy (dry), but the b and r² values in Fig. 3 show that it is not.

**Comments by Reviewer 3**

Dear authors,

I checked the authors appropriately addressed all items raised by the referee #2 except following 2 items.

(1) For the following referee's comment.

20. 5.5. The sentence is unintelligible. Moreover, the explanation of how parameter ranges (i.e. priors - why don't you call them priors) are derived is not sufficient. Provide a clear rationale for prior elicitation.

Authors replied as follows.

Parameter ranges are not priors. This is explained at the beginning of this document.

But I could not find corresponding explanation on the manuscript.

We have modified the section 2.5: "Sampling process", where the difference between the parameter ranges and priors is now explicitly stated. We also added a focus to the Table 2 (parameter descriptions) caption: "model parameters with default values, range __of acceptable values__", which should also clarify the role of the parameter ranges.

(2) On the last item from the referee#2, authors replied as follows.

The code is under MPI-M License agreement and we cannot distribute it. The driving data (approximately 500Mb) and chains can be uploaded e.g. as supplements.

For ensuring computational reproducibility, driving data, at least, should be available on an appropriate repository, which is reasonably accessible for readers.

The driving data (calibration and validation) was uploaded to the Zenodo data portal, as stated in the code availability section of the manuscript. We amended the description in the data availability section to include this information more precisely (instead of just "dataset" we state that it contains the forcing data and observations).

**Supplement S3**

This supplement is a supporting analysis of the calibration process improvements, when compared to the model with default parametrisations. The analysis is based on the slope of the regression line ($b$) and the coefficient of determination ($r^2$) from Fig. 2 and the corresponding supplementary images (S2). We calculated how many times the calibrated parameter values resulted in improvements for these variables (in boldface), how many times these values are roughly the same (the value from the default simulation is within 5 % of the corresponding value from the calibration process) and how many times the calibration has worsened the results (italic).

We urge caution in making detailed or definate conclusions based on these supporting results. This is because the Ball-Berry model and the variants are here coupled to the JSBACH model without any initial calibration. The default parameter values for these models are taken from literature – it is possible that the combination of these values and the JSBACH default parametrisation (for the other parameters) results in inferior behaviour. Therefore, comparing the calibrated results to these simulations may not be meaningful.

**Table S3a.** Model spesific analysis of the calibration process for the validation period. Improvements are given in boldface, similar behaviour without any accent and deteriorations in italic.

| mode | $b$(ET) | $r^2$(ET) | $b$(GPP) | $r^2$(GPP) | Σ |
|------|---------|-----------|----------|------------|----|
| Base | **2**,3,*5* | **9**,1,*0* | **8**,0,*2* | **8**,2,*0* | **27**,6,*7* |
| Bethy | **3**,1,*6* | **9**,1,*0* | **8**,0,*2* | **10**,0,*0* | **30**,2,*8* |
| BB | **7**,1,*2* | **10**,0,*0* | **9**,0,*1* | **9**,1,*0* | **35**,2,*3* |
| Leu | **5**,0,*5* | **10**,0,*0* | **8**,0,*2* | **8**,2,*0* | **31**,2,*7* |
| F&K | **2**,2,*6* | **10**,0,*0* | **8**,0,*2* | **10**,0,*0* | **30**,2,*8* |
| USO | **2**,2,*6* | **9**,1,*0* | **7**,0,*3* | **8**,2,*0* | **26**,5,*9* |
| Σ | **21**,9,*30* | **57**,3,*0* | **48**,0,*12* | **53**,7,*0* | |

**Table S3b.** Site spesific analysis of the calibration process for the validation period. Improvements are given in boldface, similar behaviour without any accent and deteriorations in italic. Validation site identifiers have also been italised.

[revised manuscript text omitted]

| CA-Qfo | 49.69 | -74.34 | 382 | *Picea mariana* | 3.7 | 112 | 962 | -0.4 | 2003–2010 | **?** |
| FI-Hyy | 61.85 | 24.29 | 180 | *Pinus sylvestris* | 3.5 | 45 | 709 | 2.9 | 1999–2006 | **?** |
| FI-Ken | 67.99 | 24.24 | 337 | *Picea abies* | 2.1 | 100 | 484 | 0.4 | 2003–2010 | **?** |
| FI-Sod | 67.36 | 26.64 | 179 | *Pinus sylvestris* | 1.7 | 150 | 527 | -0.4 | 2001–2008 | **?** |
| RU-Fyo | 56.45 | 32.90 | 265 | *Picea abies* | 4.5 | 200 | 711 | 3.9 | 2002–2009 | **?** |
| CA-Ojp | 53.92 | -104.69 | 579 | *Pinus banksiana* | 2.6 | 100 | 431 | 0.1 | 2004–2006 | **?** |
| FI-Let | 60.64 | 23.96 | 119 | *Pinus sylvestris* | 6.0 | 40 | 627 | 4.6 | 2010–2012 | **?** |
| RU-Zot | 60.80 | 89.35 | 121 | *Pinus sylvestris* | 1.5 | 215 | 493 | -3.3 | 2002–2004 | **?** |
| US-Prr | 65.12 | -147.49 | 210 | *Picea mariana* | 0.7 | 72 | 275 | -2.0 | 2011–2013 | **?** |

**2.2 The JSBACH model**

[revised manuscript text omitted]